# The oldest known lepidosaur and origins of lepidosaur feeding adaptations

Daniel Marke[1,2 ✉], David I. Whiteside[1,3], Thitiwoot Sethapanichsakul[1,4], Robert A. Coram[1], Vincent Fernandez[5], Alexander Liptak[6], Elis Newham[7] & Michael J. Benton[1 ✉]

The Lepidosauria is the most species-rich group of land-dwelling vertebrates. The group includes around 12,000 species of lizards and snakes (Squamata) and one species of Rhynchocephalia, the tuatara *Sphenodon punctatus* from New Zealand[1]. Squamates owe their success to their generally small size, but also to their highly mobile skull that enables them to manipulate large prey. These key features of lizard and snake skulls are not seen in *Sphenodon*, which makes it important to understand the nature of their common ancestor. Lepidosaurs originated in the Triassic 252–201 million years ago, but confusion has arisen because of incomplete fossils, many of which are generalized lepidosauromorphs, neither squamates nor rhynchocephalians[2–5]. Here we report a reasonably complete skull and skeleton of a definitive rhynchocephalian from the Middle Triassic (Anisian) Helsby Sandstone Formation of Devon, UK that is around 3–7 million years older than the oldest currently known lepidosaur. The new species shows, as predicted, a non-mobile skull but an open lower temporal bar and no large palatine teeth, and it seems to have been a specialized feeder on insects. This specimen helps us understand the initial diversification of Lepidosauria as part of the Triassic Revolution, when modern-style terrestrial ecosystems emerged.

Modern squamates owe their success to extraordinary adaptations in their skulls, which include extensive kinesis (Fig. 1) and the ability to flex the snout up and down (mesokinesis). Their skulls can also move the braincase relative to the cranium (metakinesis) and move the quadrate and jaw articulation (streptostyly). These combined joint movements enable lizards, especially snakes, to manipulate and swallow large prey. *Sphenodon* and fossil rhynchocephalians have largely akinetic skulls that are incapable of any movement except a small degree of metakinesis, which enables powerful bites but limited food manipulation. Modern *Sphenodon* and squamates differ in the presence and absence, respectively, of the lower temporal bar. In squamates, the open temporal bar is a gap between the jugal and the quadrate bones of the skull that is essential for cranial kinesis. Both *Sphenodon* and many squamates have extensive palatal dentition, including substantial teeth on the palatine bone. These palatal teeth function in various ways. The lateral tooth row on the palatine in *Sphenodon* occludes with the lower jaw teeth in shearing food, and the more centrally located palatal teeth, where present in squamates, work the food against horny growths on the tongue or grip prey in the case of snakes.

These differences between the two living clades of lepidosaurs make it hard to reconstruct the probable cranial anatomy and feeding adaptations of the ancestral lepidosaur. Indeed, it is unclear whether these early forms have a kinetic or akinetic skull and what the state of the lower temporal bar and the palatal teeth are. Knowing the ancestors of successful clades is key to dating the timing of origins and the key innovations responsible for their success.

In a review of the modern lepidosaur skull[6], predictions were made about the nature of the ancestral lepidosaur on the basis of phylogenetic retrodictions from living forms (Fig. 1) and fossils such as *Gephyrosaurus bridensis*[7]. It was suggested that the skull would have shown some metakinesis but was otherwise akinetic, and probably had an open lower temporal bar and abundant palatal teeth. In this case, rhynchocephalians retained akinesis and closed the lower temporal bar by extending and suturing the posterior process of the jugal to the quadratojugal and squamosal, as seen in modern *Sphenodon*[8]. By contrast, squamates retained the open lower temporal bar and evolved varying degrees of cranial kinesis through bone loss, bone fusion and the formation of new joints. Fossil squamates then might show varying steps along the way to full kinesis, as in many modern lizards and in snakes. All the early forms should show palatal teeth, an inheritance from earlier reptiles.

The fossil record of lepidosaurs is patchy, particularly in its older parts, for which species and specimens are rare[5,9,10]. Because of their generally small size and delicate bones, fossil lepidosaur skeletons can also be incomplete and may lack key, diagnostic characteristics. Lepidosauromorpha (or pan-Lepidosauria), the wider clade, originated in the Permian, and Lepidosauria substantially diversified in the Triassic after the end-Permian mass extinction[11]. The current oldest member of Rhynchocephalia is *Wirtembergia* from the late Middle Triassic (239–237 million years ago (Ma)), which seems to show an open lower temporal bar; however, the remains are incomplete[12]. The oldest known member of crown Squamata is *Cryptovaranoides* from the Late

[1]School of Earth Sciences, University of Bristol, Bristol, UK. [2]School of Geosciences, Grant Institute, University of Edinburgh, Edinburgh, UK. [3]Fossil Reptiles, Amphibians and Birds Section, The Natural History Museum, London, UK. [4]Department of Earth Sciences, University College London, London, UK. [5]European Synchrotron Radiation Facility, Grenoble, France. [6]Diamond Light Source, Harwell Science and Innovation Campus, Didcot, UK. [7]School of Engineering and Materials Science and Institute of Bioengineering, Queen Mary University of London, London, UK. ✉e-mail: dan_marke@hotmail.co.uk; mike.benton@bristol.ac.uk

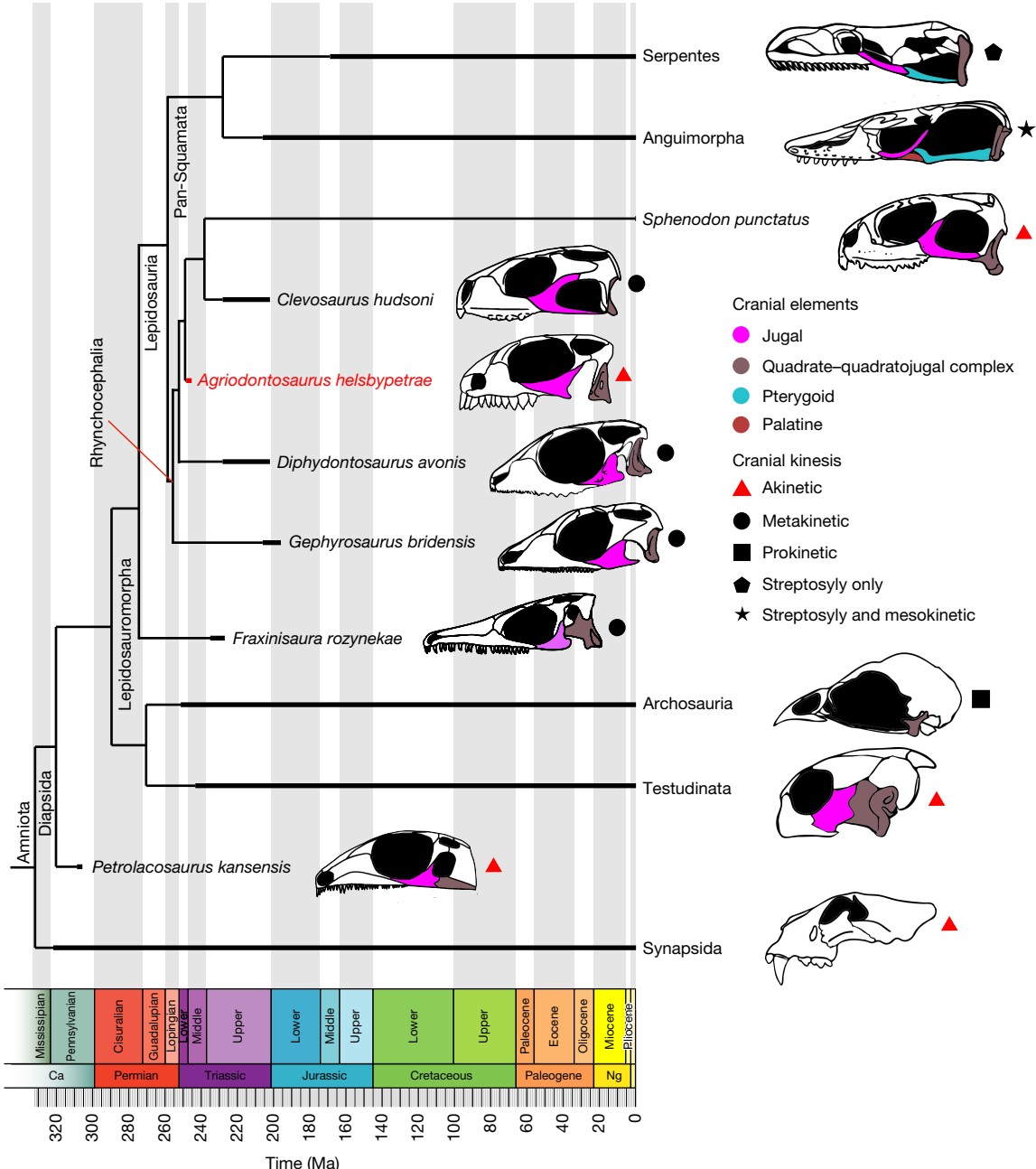

**Fig. 1 | Evolution of lepidosaurs among amniotes.** Time-scaled phylogeny illustrating the variation in morphology of the lower temporal bar and modes of cranial kinesis among fossil and extant diapsids. Skulls are not to scale. Ca, Carboniferous; Ng, Neogene.

Triassic (202 Ma), a squamate with anguimorph affinities that has an open lower temporal bar and limited cranial kinesis[13,14].

Here we present a new specimen, the oldest known member of Lepidosauria, that sheds light on the early history of Lepidosauria and of Rhynchocephalia in particular. The specimen confirms that the open lower temporal is the original condition in Lepidosauria. Moreover, mesokinesis and streptostyly were absent and, unexpectedly, palatal teeth were also absent.

## Systematic palaeontology

<div align="center">

Lepidosauria Haeckel, 1866
Rhynchocephalia Günther, 1867
Sphenodontia Williston, 1925
*Agriodontosaurus* gen. nov.
</div>

**Type and only known species.** *Agriodontosaurus helsbypetrae* gen. et sp. nov.

**Etymology.** *Agrio* from the ancient Greek epithet of Dionysus, Agrionius, meaning 'fierce' and *donto* for 'tooth', which refers to the remarkably large teeth on parts of the dentary and maxilla, and *saurus* for 'lizard'. Therefore, 'fierce-toothed lizard'. The specific term '*helsbypetrae*' refers to the Helsby Sandstone Formation (locally called the Otter Sandstone), the deposit in which the fossil was found; *petrae* is the genitive of petra, the latinized form of the ancient Greek word for rock.

**Holotype.** BRSUG 29950-14 (Fig. 2 and Extended Data Fig. 1), a partial skeleton comprising cranial and postcranial elements. The skull is distorted and lacks the rostrum; the left side is more complete than the right side of the skull. Parts of the palate and braincase are damaged. The skull is estimated to have been about 14 mm in length. Also preserved are postcranial elements, including an articulated sequence of cervical and dorsal vertebrae, articulated pectoral and disarticulated

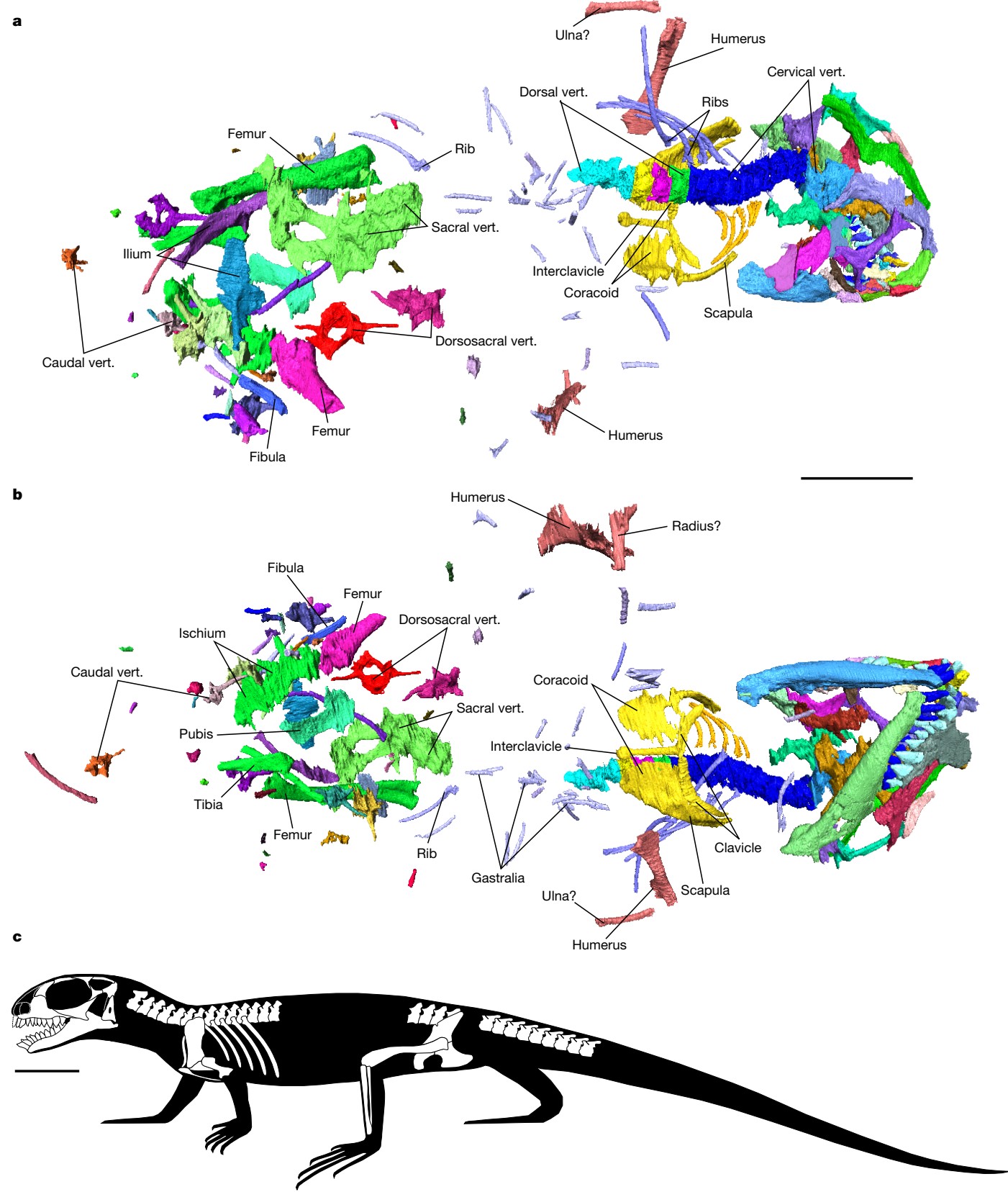

**Fig. 2 | Holotype specimen of *A. helsbypetrae* holotype BRSUG 29950-14. a,b**, Three-dimensional reconstruction of the skeleton (voxel resolution of 26 μm; Methods) in dorsal (**a**) and ventral (**b**) views. **c**, Lateral reconstruction of *A. helsbypetrae* based on elements preserved in the holotype. Scale bars, 10 mm.

Full details are provided in Extended Data Figs. 4–9. Colours denote individual segmentations of elements, or groups of elements that were segmented together. vert., vertebrae.

pelvic girdles, several dorsosacral and caudal vertebrae and the proximal limb bones.

**Locality and age.** The specimen is from the Helsby Sandstone Formation of Sidmouth, Devon, UK. It was excavated as a block by R.A.C. in 2015 from a temporarily exposed foreshore exposure beneath Peak Hill (UK National Grid Reference SY 109865), from the upper half of the formation, perhaps upper Anisian (244–241.5 Ma)[15].

**Diagnosis.** Small rhynchocephalian with a body length of about 100 mm with a unique combination of the following 13 features: dentary and maxillary anterior teeth are simple and conical but robust; posterior teeth are more triangular with broad bases and set slightly en echelon; anterior maxillary and dentary teeth are acrodont and posterior teeth are pleuracrodont with a residual subdental shelf; maxilla with a pronounced anterior process and high facial process; lateral tooth row on the palatine absent; broad, flat parietal table composed of paired bones; ventral region of orbit bounded mainly by the jugal, which provides about 90% of the boundary and the maxilla the rest; jugal with a prominent, but short, posteroventral process that does not reach halfway in the ventral region of the lower temporal opening; quadrate with conch and large foramen; dentary extends posteriorly to underlie the glenoid of the lower jaw; fused prearticular, articular and surangular in the lower jaw; bicapitate ribs in the cervical and trunk regions; gastralia present; and bulb-shaped expansion of the posteriormost part of interclavicle.

## Morphological description

### Cranium

The anteriormost skull bones, the premaxillae, nasals and putative septomaxillae, are not preserved. The maxillae (Fig. 3c–e and Extended Data Fig. 2a–d) resemble the morphology in the basal rhynchocephalians *Diphydontosaurus*, *Planocephalosaurus* and *Gephyrosaurus*[16], as well as in the extant *Sphenodon*. Specifically, the maxillae possess pronounced facial and anterior processes that form the margins of the external nares. The medial surface of the maxillae bears supradental shelves that support the maxillary teeth (Extended Data Fig. 2d,k). When viewed laterally, a facet for articulation of the jugal (Extended Data Fig. 2a) can be seen where it limits, but does not exclude, a maxillary contribution to the ventral margin of the orbit (Fig. 3d).

Both prefrontals are preserved, but the left and right elements lack their dorsal and ventral portions, respectively (Fig. 3a–e,h). Although the ventralmost extent of the left prefrontal is not preserved, we surmise that it would have contacted the palatine to brace the skull roof, a diagnostic characteristic of lepidosaurs[17]. The facet is highly probably marked dorsally by a raised area on the anterior side of each palatine, medial to the maxillary facet. Lacrimals and associated facets on the maxilla are absent, which is the case in many rhynchocephalians, particularly sphenodontians[16,18,19].

The frontals are paired, delineated by a midline suture (Fig. 3a), and together they form an hourglass shape in dorsal view. When viewed ventrally, there is a pronounced thickening of the lateral margins of the frontals that forms the olfactory process along the entire dorsal margin of the orbit. The parietals are paired, and despite abrasion, there is a notable thinning towards the midline that would not be expected if the parietals were fused. There is evidence of a pineal foramen (Fig. 3a,h). The left parietal bears a posterior process, curving to contact the squamosal and form the margin of the supratemporal fenestra (Fig. 3a,h).

The postfrontals form a lunate contact at the frontoparietal suture, a feature common in stem lepidosaurs[17], and contribute substantially to the margins of the supratemporal fenestra. The postorbitals (Fig. 3a,c–h) are the typical triradiate form of lepidosaurs[16] and contribute to much of the ventral margin of the supratemporal fenestra (Fig. 3d).

Only the left jugal (Fig. 3d,e) is preserved, with an anterior process that overlaps the maxilla and forms approximately 90% of the ventral margin of the orbit. There is no contact with the squamosal as in *Sphenodon* (Fig. 3d,e), but unlike basal rhynchocephalians such as *Planocephalosaurus*[20] and clevosaurs[21,22]. There is a distinct and spur-like posteroventral process, which indicates an incomplete lower temporal arcade (Fig. 3d,e) that is more reduced than in stem-lepidosaurs and in rhynchocephalians such as *Gephyrosaurus*, *Diphydontosaurus*, *Planocephalosaurus*, *Clevosaurus* and *Palaeopleurosaurus*[16,23]. However, it is more pronounced than in the pan-lepidosaurs *Marmoretta* and *Paliguana*[11].

The left squamosal is poorly preserved (Fig. 3a,d,e,h), but the outline confirms the typical tetraradiate shape of early diapsids and rhynchocephalians[5]. There is no contact with the jugal, but the fragmentary right squamosal is attached to the cephalic head of the right quadrate, holding the bone firmly, thereby confirming that the quadrate was akinetic.

Both quadrates are present, although the right is significantly more complete (Fig. 3a,b,e and Extended Data Fig. 3a–d). The ventral end of the quadrate shows a distinct notch-like termination and the condyles for articulation with the articular (Fig. 3a,b,e and Extended Data Fig. 3a). The anterior and posterior margins of the quadrate are thickened, surrounding a central fossa with a prominent quadratojugal foramen and a weakly developed lateral conch (Fig. 3a,e and Extended Data Fig. 3a) that is less developed than in basal rhynchocephalians[6,8,22,24] and the lepidosauromorph *Sophineta*[2]. The anterior margin bears two probable facets to receive the ventral process of the squamosal (Extended Data Fig. 3a), which means that the squamosal may have extended as far ventrally as the putative quadratojugal, although damage to the area prevents confirmation of this possibility. From computed tomography (CT) slices (Supplementary Data 1 and 2), we identified that a broad flange connected the quadrate and pterygoid medially, which is a typical feature of rhynchocephalians[6].

The quadratojugals are indistinguishable from the quadrates. The right quadratojugal may be represented by a bulbous expansion of the anteroventral margin of the right quadrate, immediately dorsal of the anterior condyle (Fig. 3a,e and Extended Data Fig. 3a,b) and bordering the tympanic crest.

### Palate and braincase

Most sutures between elements of the palate are obscure and cannot be discerned in the specimen or in the CT scans, partially because of skull deformation. Only part of the right vomer is preserved (Fig. 3i and Extended Data Fig. 3e,f) and is flat in cross-section, demarcated from the anterior of the right palatine by a ridge on the CT model.

The palatines (Fig. 3i and Extended Data Fig. 3e,f) bear maxillary processes that form the anterior margin of a distinct suborbital fenestra, delineated posteriorly by the anterior edge of the ectopterygoid (Fig. 3i and Extended Data Fig. 3e,f). Both palatines also lack any evidence of enlarged teeth (Extended Data Fig. 3e,f), and CT slices (Supplementary Data 1 and 2) show that they are absent, but there are some features that might be the bases of smaller denticles. We devoted much time to searching for palatal teeth on the scans and found no evidence of them. Their absence is further supported by the lack of tooth-wear facets on the lingual side of the dentary (Extended Data Fig. 2g,h), an indicator of the occlusion of the margins of palatine teeth as seen in many rhynchocephalians such as the clevosaurs. Large palatine teeth are about half the size of the maxillary teeth and conspicuous in the palates of *Clevosaurus*[24], *Planocephalosaurus*[20] and *Diphydontosaurus*[8]. Therefore, their apparent absence in *Agriodontosaurus* was unexpected.

The pterygoids (Fig. 3i and Extended Data Fig. 3e,f) also show no evidence of dentition, which is unique among early lepidosaurs, including rhynchocephalians, and is seen only in the derived *Priosphenodon*[25] and *S. punctatus*. The knobbed structures are an artefact of segmentation, and no definitive evidence of dentition was visible in the scan. The medial margins of each pterygoid bound large fossae for the basipterygoid processes (left basipterygoid process; Extended Data Fig. 3e).

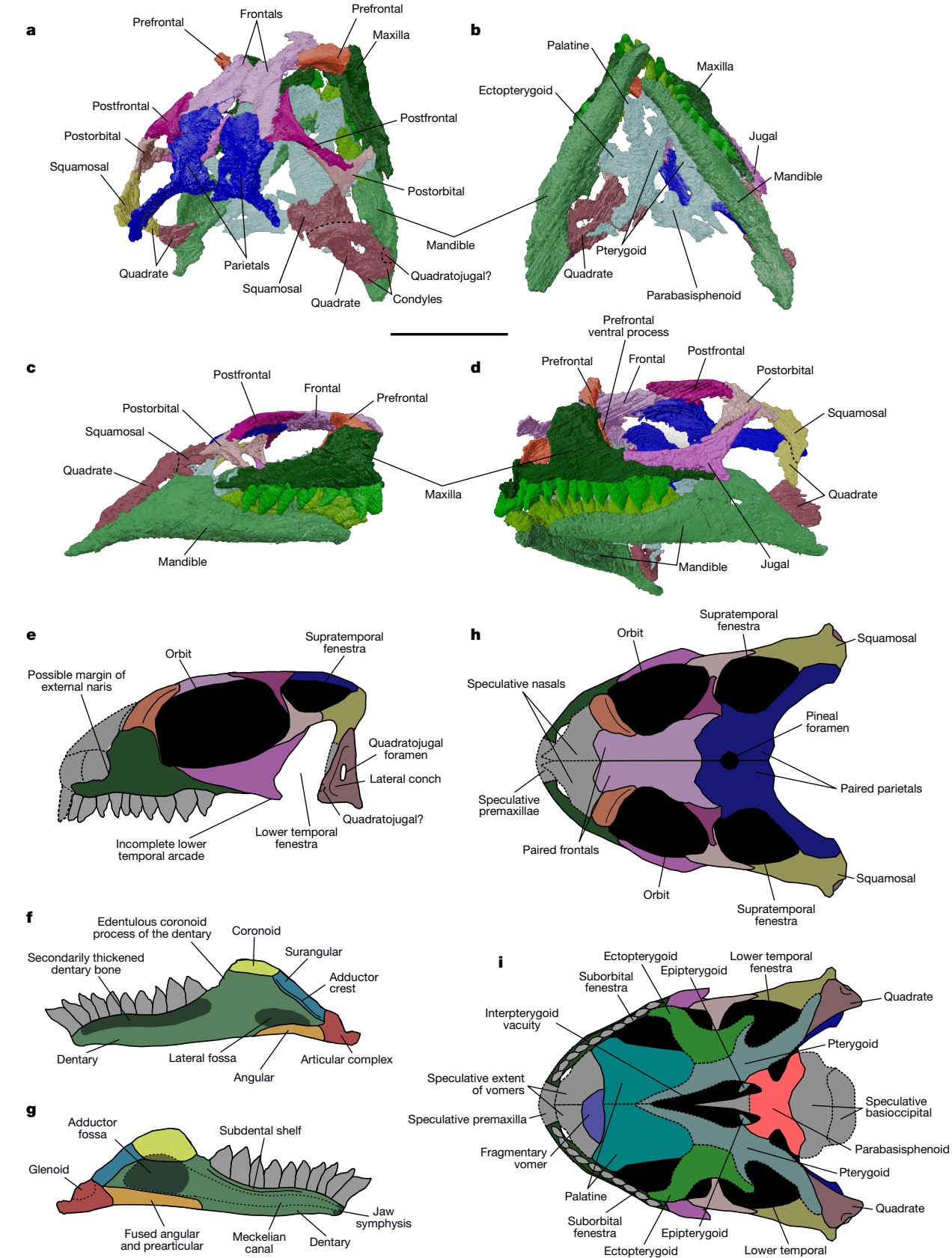

**Fig. 3 | Skull of holotype specimen *A. helsbypetrae* holotype BRSUG 29950-14.**
**a**–**d**, Three-dimensional reconstruction of the skull of *A. helsbypetrae* holotype (voxel resolution of 6 μm; Methods) in dorsal (**a**), ventral (**b**), and right (**c**) and left (**d**) lateral views. **e**–**i**, Reconstructed skull and mandible of *A. helsbypetrae* in lateral (**e**,**f**), medial (**g**), dorsal (**h**) and ventral (**i**) views. Bones in grey and contacts marked with dashed lines are unknown and speculatively reconstructed. Scale bar, 5 mm. Full details of tooth-bearing bones, right quadrate and palate are provided in Extended Data Fig. 3. Colours indicate paired elements between the segmentations and the illustrated reconstructions.

The main body is flat, differing from squamates that bear a distinct concavity bound by the lateral margins[5].

The ectopterygoid is broad as it extends anterolaterally—to contact the posterior process of the maxilla—and posteriorly as it forms an arcuate flange with the pterygoid transverse process (Fig. 3i and Extended Data Fig. 3e,f), as seen in *Clevosaurus hudsoni* and *Clevosaurus cambrica*[21]. This posterolateral excavation probably accommodated the m. adductor mandibularis.

The dorsal surface of the pterygoid near the right basipterygoid process shows a clear outline and contact with the lower part of the epipterygoid (Extended Data Fig. 3e). Its position precludes any contact with the quadrate anterior lamella. Its base is more like the broad shape in *Sphenodon* than the more columnar form in *Clevosaurus*[26].

Very little of the braincase is fossilized. The parabasisphenoid (Fig. 3i and Extended Data Fig. 3e,f) is poorly preserved, but the posterior extent of its palatal ramus is bordered laterally by both basipterygoid processes. These are short and thick, and the ventral region where it articulates with the pterygoid pocket is visible only in the left basipterygoid (Extended Data Fig. 3f) where it strongly resembles that in *Sphenodon*[27]. The main body of the parabasisphenoid bears a single ventral concavity (Extended Data Fig. 3f) like that in other rhynchocephalians such as *Diphydontosaurus*[8] and *Sphenodon*.

## Lower jaw

The mandible (Fig. 3f,g and Extended Data Fig. 2e–j) is robust and each hemimandible possesses fewer teeth (around 10) than early pan-lepidosaurians (for example, 25–30 in *Sophineta*[2]). The dentaries make up most of the lower jaw length (about 85%) and are complete with the full sequence of teeth on the right dentary and a partial sequence on the left. As in sphenodontians, the upper mid-lateral region of the dentary has a secondary thickening of bone (Fig. 3f and Extended Data Fig. 2e) that runs longitudinally. However, no tooth-wear facets into the bone can be discerned. The unfused symphyses are almost vertical, with a rough articulating contact (Extended Data Fig. 2h) that is divided medially by the Meckelian canal into a dorsal and ventral facet (Fig. 3g and Extended Data Fig. 2g,h). The Meckelian canal extends along the entire medial face of the dentary, delineated dorsally by thickened bone that bears a pronounced subdental shelf that supports the dentary teeth (Fig. 3g and Extended Data Fig. 2g,h). This feature is common in basal rhynchocephalians such as *Deltadectes*, *Diphydontosaurus* and *Planocephalosaurus*[8,28]. A single fused posterior process projects as far as the articular condyles (Fig. 3f,g and Extended Data Fig. 2e–h). There is a small edentulous section that lies at the base of the pronounced coronoid process, which is overlapped by the coronoid. The coronoid process, when compared with the dentary thickness below the mid tooth row, is dorsally expanded far beyond the extent seen in *Wirtembergia*[12] and *Diphydontosaurus*[8]. However, it is not as high or pointed as in *C. hudsoni*, *C. cambrica*[21] or *Clevosaurus brasiliensis*, more resembling *Microsphenodon*[29] and *Sphenodon*[6]. The splenials are absent, with no splenial facets on the dentaries to accommodate them. We therefore suggest that they were absent in life, a condition shared with other rhynchocephalians[18]. Where the dentary meets the posterior bones of the lower jaw, the Meckelian canal expands into a large medial fossa for attachment of the m. adductor mandibularis (Fig. 3g and Extended Data Fig. 2g,h).

The sutures between the post-dentary bones, except the coronoid, are difficult to discern but we identified probable contacts from landmarks in the CT surface models (Fig. 3f,g and Extended Data Fig. 2e–h). Like *C. brasiliensis*, *Microsphenodon*, *Sphenodon* and indeed rhynchocephalians in general, the coronoid is more extensive on the medial than the lateral side of the coronoid process (Fig. 3f,g and Extended Data Fig. 2e–h). It is also strongly concave, contributing heavily to the medial fossae to which the m. adductor mandibularis would attach. There is no anterolateral (that is, labial) process, but medial processes are uncertain and are not coded.

On the basis of recognizable ridges on the lateral surfaces of the mandible, we surmise that the angulars underlie the posterior process of the dentary, which makes them only visible as small slivers in lateral view, expanding from below the coronoid process as far posteriorly as the articulars (Fig. 3f,g and Extended Data Fig. 2g,h). The left surangular slightly overlaps the posterior surface of the coronoid process and contributes modestly to its dorsal eminence. Posterolaterally, the face of the surangular is thick, forming a pronounced lateral adductor crest and lateral fossa (Fig. 3f and Extended Data Fig. 2e,f). The articulars (Fig. 3f,g and Extended Data Fig. 2e,j) are marked by the glenoid, of which the lateral margin is higher than the medial, forming a posteromedially oriented facet. At the posterior end, there is a distinct retroarticular that tapers posteriorly into a hook-like process.

## Dentition

There are up to ten teeth on the dentary and ten on the maxilla (Fig. 3e–g and Extended Data Fig. 2a–j). All teeth are pointed, but the anterior teeth are simple cones with large pulp cavities that differ from the posterior. These teeth have distinct broader crowns and are more triangular, with smaller pulp cavities and are labiolingually compressed (Extended Data Fig. 2i–k) and mesiodistally elongated. There are examples of smaller teeth in the anterior series, which may indicate replacement. Posterior teeth are set en echelon with the mesial aspect of a maxillary tooth partially labially overlapping its anteriorly positioned neighbour (Fig. 3a,c,i and Extended Data Fig. 2i,j), and there is a similar partial overlap on the dentary dentition. The maxillary and dentary posterior teeth each have a posterolingual flange but lack mesial flanges (Fig. 3f,g and Extended Data Fig. 2e–h), which are present in clevosaurs such as *C. hudsoni*[21,24]. The shape and positioning of the posterior teeth permitted pronounced shear during occlusion, as indicated by tooth-wear facets and the CT slices (Supplementary Data 1 and 2).

The teeth are very large in proportion to the tooth-bearing marginal bones, with many as high as, or higher than, the dorsoventral thickness of the bone to which they are attached. The posterior teeth are ankylosed to distinct supradental/subdental shelves, which are seen medially, and the labial apex of each tooth-bearing marginal bone (pleuracrodonty, sensu[28]) (Fig. 3g and Extended Data Fig. 2c,d,g,h,k–i). Anteriorly, the teeth are ankylosed directly to the apex of the tooth-bearing marginal bones (acrodonty) (Extended Data Fig. 2i,j). The largest teeth of the tooth-bearing bones are the most mesial or penultimate distal teeth. The shape and mode of implantation of the teeth support a rhynchocephalian affinity, being primarily acrodont as in *Clevosaurus*, *Sphenodon* and others, even though *Wirtembergia* and *Diphydontosaurus* have pleurodont anterior teeth and acrodont posterior teeth. The exaggerated anterior teeth are rare among early representatives of the clade, but this suite of features can be observed in clevosaurids such as *Clevosaurus hadroprodon*[30].

The postcranial skeleton is described in the Supplementary Information and Extended Data Figs. 4–6.

## Phylogenetic analysis

Parsimony-based phylogenetic analyses[5,29] resolve *A. helsbypetrae* as a sphenodontian rhynchocephalian positioned crownward of *Gephyrosaurus* and *Diphydontosaurus*. Bayesian phylogenetic analysis (Fig. 4) clusters *A. helsbypetrae* with *Wirtembergia*, *C. hadroprodon* and *Parvosaurus* (posterior probability = 0.50), positioned crownward of *Gephyrosaurus* and *Diphydontosaurus* but basal to *C. hudsoni* and *Planocephalosaurus*. Furthermore, the procolophonids *Kapes* and *Procolophon* form a strongly supported clade along with *Palacrodon* (posterior probability = 1.0) at the base of the phylogeny, whereas *Sophineta* is resolved as a pan-lepidosaur (Fig. 4 and Extended Data Fig. 7). Assessed against a more diverse dataset of rhynchocephalians[29], parsimony analysis produced 272 most parsimonious trees of 713 steps, a 50% majority rule and strict consensus of which resolves

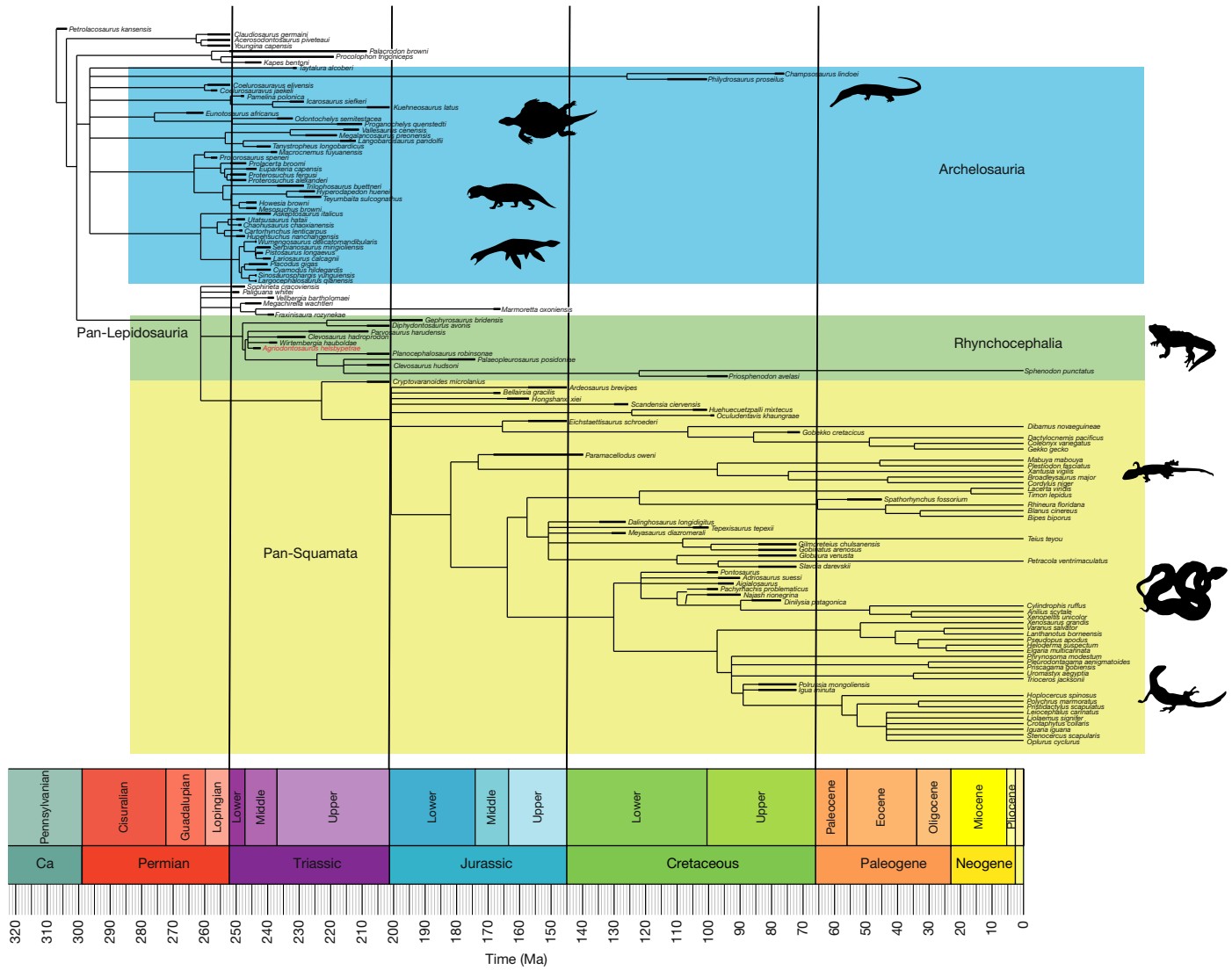

**Fig. 4 | Phylogenetic analysis of Diapsida and position of *A. helsbypetrae*.** Maximum parsimony tree from a Bayesian-inference analysis of data modified from the data matrix of ref. 5, plotted against geological time using ruta rooting. Silhouettes were obtained from Phylopic (https://www.phylopic.org). *Champsosaurus*, created by S. Traver under a CC0 1.0 Universal Public Domain licence; *Odontochelys*, created by T. M. Keesey under a CC0 1.0 Universal Public Domain licence; *Hyperodapedon*, created by S. Traver under a CC0 1.0 Universal Public Domain licence; *Morturneria*, created by A. S. Brum under a CC0 1.0 Universal Public Domain licence; *Sphenodon*, created by S. Traver under a CC0 1.0 Universal Public Domain licence; *Gekko*, created by S. Traver under a CC0 1.0 Universal Public Domain licence; *Calloselasma*, created by S. Traver under a CC0 1.0 Universal Public Domain licence; *Varanus*, created by S. Traver under a CC0 1.0 Universal Public Domain licence.

*Agriodontosaurus* crownward of *Diphydontosaurus*, *Planocephalosaurus* and *Parvosaurus* with strong support (node frequency = 1) (Extended Data Figs. 8 and 9).

Our analyses suggest that *Sophineta*, *Vellbergia*, *Paliguana*, *Megachirella*, *Marmoretta* and *Fraxinisaura* are pan-lepidosaurs rather than stem squamates or stem rhynchocephalians as proposed by some studies[13,31,32], but consistent with other studies[5,13,33]. *Cryptovaranoides* is resolved variously as a sister to crown Squamata or in crown Squamata (Fig. 4 and Extended Data Figs. 7–9), but the data matrix used here lacks definitive characteristics that ally this taxon with Toxicofera and Anguimorpha[13,14].

## Discussion

*Agriodontosaurus* is one of the earliest occurring sphenodontian rhynchocephalians, being 3–7 million years older than the next oldest rhynchocephalian: *Wirtembergia* from the Erfurt Formation (late

Ladinian, 238 Ma). This finding means that the key phylogenetic split in Lepidosauria, between Rhynchocephalia and Pan-Squamata, occurred minimally at 245–241 Ma. This timeline was not unexpected according to some phylogenies that include taxa such as *Megachirella* (245 Ma)[31], *Fraxinisaura* and *Vellbergia* (both 238 Ma)[4,34] as stem squamates, but we found that these fall outside the clade Lepidosauria. Therefore, currently, *Agriodontosaurus* is both the oldest known member of Rhynchocephalia and of Lepidosauria. Furthermore, it provides support for a European origin for Lepidosauria[23,35].

Fixing dates of the oldest fossil members of crown clades is crucial for dating the tree of life[36–39], and clarity is required over which are members of crown clades or total clades (=crown + stem). Fossils outside the crown cannot contribute to estimating origin dates of modern taxa. *Agriodontosaurus* fixes a minimum age constraint on the origin of Lepidosauria at 245–241 Ma. The maximum age, or lower bound, is harder to fix. Some studies[31,40] project origins of major diapsid clades, including Lepidosauria, into the Permian at around 260 Ma, long before

the occurrence of the first fossils. It is unclear, however, whether Lepidosauria had much of a Permian history, but they probably expanded substantially after the end-Permian mass extinction 252 Ma as part of the Triassic Revolution[23,41].

Our conclusion that *Agriodontosaurus* belongs to crown Lepidosauria is supported by the following apomorphies[17]: prefrontals (probably) bracing skull roof on the palate; lacrimal bones largely confined to the orbital rim (in this case, they are entirely absent); marginal teeth attached superficially to the lingual surface of posterior parts of the jaw (rather than in shallow sockets, further modified to more apical attachment in some taxa); and teeth lost from the pterygoid transverse process and from sphenoid bones. *Agriodontosaurus* also possesses two unexpectedly derived features: few or no teeth on the palatal bones and proportionally large teeth on maxillae and dentaries. The incomplete lower temporal bar in early rhynchocephalians was predicted[8], and the mix of acrodont and pleuracrodont marginal teeth is found in basal sphenodontians such as *Diphydontosaurus avonis*. However, the fully acrodont teeth are anterior in *Agriodontosaurus* rather than posterior as in *Diphydontosaurus*. The quadrate conch and prominent olfactory process of the frontal are said[11] to distinguish Lepidosauria from Pan-Lepidosauria; both are present in *Agriodontosaurus*, in which the olfactory processes are similar to those of *Clevosaurus*[42]. The posteriorly bowed quadrate of *Agriodontosaurus* is also a pan-lepidosaurian character[11].

We also identified the following apparent squamate apomorphies in *Agriodontosaurus*: jugal closely approaches prefrontal and exposed above the orbital margin of maxilla, mostly excluding the latter from the ventral margin of the orbit; and quadratojugal absent as a discrete element[17]. Conversely, the suborbital process of the jugal that forms most or all of the ventral margin of the orbit has been regarded as a plesiomorphic feature as it is found in *Paliguana*[11].

It was previously predicted[6] that the common ancestor of lepidosaurs would possess the following features: paired parietals with a large central parietal foramen; parietal table roughly equal in length to postparietal processes; frontal is the longest skull roofing bone; strongly interdigitated W-shaped frontoparietal suture braced by a strong suture with the postfrontal; tall facial process of the maxilla; postorbitals and postfrontals large and separate, with both forming margins of the orbit and upper temporal fenestra; jugal triangular with barely any posterior process; lateral head of ectopterygoid small and does not contact palatine; large quadrate with a lateral conch firmly held by the pterygoid and squamosal; and squamosal attached to the parietal. All these features are found in *Agriodontosaurus*.

However, *Agriodontosaurus* differs from the previous predictions[6] in the following ways: lack of contact between jugal and squamosal (but similar to *Paliguana*[11]); a robust, not long and thin, dentary; and the squamosal, although quadriradiate, is more gracile than predicted. Moreover, although the maxilla contacts the orbital margin, it is less extensive than envisaged, with the jugal positioned well above the maxilla and extending a long way anteriorly (more akin to the squamate condition). Furthermore, there is no evidence of a supratemporal, the pterygoids probably did not contact the vomers (we considered that the palatines contacted in the midline) and, most notably, there is little evidence of a palatal dentition.

Some of these differences from predicted basal lepidosaurian features[2] are because *Agriodontosaurus* is a rhynchocephalian. Similarly, the lack of a splenial and small angular and particularly the presence of acrodont or pleuracrodont dentition are characteristic of the Rhynchocephalia. *Agriodontosaurus* has the short, dorsoventrally thick, dentary with a large coronoid process found in most rhynchocephalians. The acrodont dentition does not contradict the hypothesis that the basal lepidosaur had pleurodont teeth, as the sub and supra-dental shelves in *Agriodontosaurus* demonstrate derivation from pleurodonty. The basipterygoid processes are short and thick like those of *Sphenodon* and, like that taxon, probably limited

metakinesis, which therefore leaves its rhynchocephalian skull largely akinetic[43].

*Agriodontosaurus* differs from later Triassic rhynchocephalians such as *Gephyrosaurus* and *Diphydontosaurus* in having a less robust posterior portion of the skull and a relatively larger lower temporal fenestra, although the lower temporal area is similar in magnitude to clevosaurs. *Agriodontosaurus* also has a prominent anterior process of the maxilla. This feature is found in stem lepidosaurs such as *Marmoretta* and *Fraxinisaura*[33] and the pleurodont rhynchocephalian *Gephyrosaurus* but not in the acrodont clevosaurs, although a similar feature, somewhat less developed, is found in *Sphenodon*.

The acrodont–pleuracrodont dentition of *Agriodontosaurus* is intermediate in some ways between that of stem-lepidosaurs and rhynchocephalians (typically pleurodont or subpleurodont) to that of early squamates (pleurodont-only) and derived rhynchocephalians (acrodont).

The probable absence of palatal teeth in *Agriodontosaurus* contrasts with the numerous palatal teeth of clevosaurs[29] and the rhynchocephalians *Gephyrosaurus*, *Diphydontosaurus* and *Planocephalosaurus*. The Ladinian rhynchocephalian *Wirtembergia* has pterygoid teeth in two rows and there are other small palatal teeth. Some stem lepidosaurs such as the mid-Jurassic *Marmoretta*[33] also have numerous palatal teeth, but the palate is poorly known in Early Triassic pan-lepidosaurs such as *Paliguana*[11] and *Sophineta*[2]. However, recent finds of Triassic lepidosauromorphs, including the Ladinian *Fraxinisaura*[34] and Late Carnian *Taytalura alcoberi*[32], are reconstructed with far fewer palatine teeth or pterygoid teeth than, for example, the Late Triassic or Early Jurassic *Gephyrosaurus* or *Diphydontosaurus*. Palatal teeth re-evolved in Squamata[44,45], and we propose that the same occurred in Rhynchocephalia. Perhaps the row of large lateral palatine teeth co-evolved with the development of a more posteriorly extending jugal posteroventral process. Unfortunately, the palatine is unknown or represented only by scraps of bone in *Sophineta*[2], *Vellbergia*[4], *Megachirella*[31], *Wirtembergia*[12] and *C. hadroprodon*[30].

The absence of a lower temporal bar in early lepidosaurs and squamates has long been acknowledged as the ancestral condition for Lepidosauria[8,16,23] and was reacquired in *Priosphenodon*[25] and *Sphenodon*[8]. The distinct posteroventral process of the jugal in *Agriodontosaurus*, although less developed than in *Gephyrosaurus* and *Diphydontosaurus*, could indicate that rhynchocephalians had begun to reacquire the lower temporal bar earlier than had been thought[7,12]. However, the less developed posteroventral process in *Wirtembergia*[12] suggests that the lower temporal bar might have re-evolved more than once among Rhynchocephalia. Although the posteroventral process of the jugal is distinct, the quadratojugal is barely distinguishable and probably fused to the quadrate, unlike in *Gephyrosaurus* or *Diphydontosaurus*, which have well-developed quadratojugals. Reacquisition of the lower temporal bar might initially have occurred with development of the posteroventral process of the jugal and the quadratojugal perhaps enlarging in later taxa such as *C. hudsoni*[22].

The anatomy of *Agriodontosaurus* provides evidence about its ecology and function. The unique dentition and skull suggest that this was an active predator on large insects, a strategy of prey capture not seen in other Triassic stem or crown lepidosaurs. Evidence that this animal could rapidly deliver a substantial bite force are the dorsally expanded coronoid processes of the dentary, the thick lateral crested rims of the parietal table, the adductor fossae with the large open lower temporal fenestrae and the large upper temporal openings. Furthermore, the position of the glenoid ventral to the tooth row and short retroarticular would enable significant gape capability and enable the teeth to occlude simultaneously along the tooth row[16]. The anterior large conical piercing teeth would also prevent any forward movement of the prey during occlusion, and the quadrate held securely by the squamosal would have facilitated a firm articulation of the condyles with the glenoid of the lower jaw.

These adaptations enabled *Agriodontosaurus* to hold its struggling prey firmly while it pierced the exoskeleton with its conical and pointed tooth crowns. At the same time, it could shear the tissues and divide the prey into manageable chunks through occlusion of the wide, triangular offset posterior dentition[16]. There are labial wear facets on the mid mandibular teeth, which demonstrates that the lower and upper teeth cut against each other. Overall, the substantial marginal dentition might have reduced the need for palatal teeth in prey prehension and perhaps suggest a reason for their absence in *Agriodontosaurus*. We also speculate that, like *Sphenodon*, *Agriodontosaurus* was a lingual feeder that used its tongue to manipulate prey in the mouth, placing it between the large marginal teeth.

We suggest that *Agriodontosaurus* had keen senses of sight and hearing on the basis of the relatively large orbits and the presence of the lateral conch on the quadrate, necessary for hunting fast-moving prey[8,16]. Arthropods available in the Middle Triassic, all of which might have formed parts of the diet of *Agriodontodaurus*, included polyneopteran insects such as dictyopterans (cockroaches and relatives) and orthopterans (grasshoppers and crickets)[46].

The tooth shear in *Agriodontosaurus*, despite representing a hard bite with maxillary and dentary teeth cutting against each other, lacks the precise slicing found in the later Triassic *Diphydontosaurus*[8] and particularly in clevosaurs. There are wear facets below the mid and posterior teeth on the lateral side of the dentary in *Wirtembergia*[12] caused by upper teeth cutting against the dentary. In the clevosaurs, the teeth are self-sharpening[16,21] as they cut deeply against each other in occlusion. The deep wear facets formed in the lateral side of the dentary in *C. hudsoni* by this precise shear[21] are not observed in the CT scans of *Agriodontosaurus*. The clevosaur postorbital bar is more robust and the overall postorbital region of clevosaurs, with the well-developed posteriorly elongated jugal process, comprises a greater proportion of the skull[16] than in *Agriodontosaurus*. Therefore, evolution towards a stronger skull structure, including a more complete lower temporal bar, to withstand the forces developed during occlusion separates these Middle and Late Triassic taxa.

The suite of features observed in *Agriodontosaurus* and our identification of it as a crown lepidosaur confirms that the Triassic Revolution was a time of rapid diversification of early Lepidosauria, pushing the origins of several features of derived rhynchocephalians and squamates back to at least the upper Anisian (245–241 Ma). For example, although the skull is essentially akinetic, except for a possible small amount of metakinesis, it shares some features of squamates such as the jugal forming most of the ventral margin of the orbit and the absence of the lower temporal bar. The absence of the lower temporal bar does not equal kinesis, but it is a prerequisite for later evolution of streptostyly and mesokinesis.

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

## Methods

### Locality and geological setting

The Otter Sandstone, formerly the Otter Sandstone Formation, is an informal term to refer to the Middle Triassic red sandstones of Devon, southern UK, assigned now to the Helsby Sandstone Formation (Sherwood Sandstone Group)[47]. Magnetostratigraphical data[48] indicate that the Otter Sandstone is Middle Triassic (Anisian) in age and that the top of the sequence is close to the Anisian–Ladinian boundary, 242 Ma. The formation comprises up to 200 m of mostly fining-upwards, reddish-hued sandstones separated by thinner layers of conglomerates and mudstones, deposited by a migrating river system under a semi-arid paleoclimate. Fossils include plants, freshwater bivalves, fishes, temnospondyls, procolophonids, rhynchosaurs and archosaurs, all occurring in the upper half of the formation, so probably dating to the upper half of Anisian time[15,49]. Vertebrate remains from the Otter Sandstone are preserved both as abraded bones and bone pebbles in channel lags and occasionally as more complete material, usually in overbank deposits.

A recently discovered thin (about 20–40 mm) sandstone bed located near the top of the Otter Sandstone (Pennington Point Member[50]) on the coast west of Sidmouth, east Devon, has produced an abundance of vertebrate fossils[49]. Among these often remarkably well-preserved remains is the nearly complete skull and skeleton of a rhynchocephalian, described here, which was excavated as a block by R.A.C. in 2015 from a temporarily exposed foreshore exposure beneath Peak Hill (National Grid Reference SY 109865). This bone bed was also the source of nearly complete skeletons of fishes, as well as articulated remains of the procolophonid *Kapes bentoni*[51] and the small neodiapsid *Feralisaurus corami*[52]. The quality of preservation and articulation of these fishes and small reptiles suggests that they had not been transported far, and there is evidence that they were living on the ponded surface of an exposed channel bar when an adjacent river channel overtopped its banks, probably after a severe rainstorm[49]. This event rapidly tumbled and then entombed the smaller inhabitants in sediment.

### Specimen

The specimen, BRSUG 29950-14, is remarkably preserved and partially articulated, with some minor damage to peripheral structures. The skeleton is contained in a block of fine-grained sandstone measuring 105.9 × 123.7 × 43.8 mm. Most of the skeleton is embedded roughly 15 mm below the surface of the block, and the dorsal margins of the skull, several vertebrae, ribs, pelvic elements and hind limb bones are visible on the surface. These were exposed partly by natural weathering before collection and then by exploratory physical preparation of the specimen by R.A.C. The specimen was soaked for several days in fresh water soon after collection to remove impregnated salt.

### Institutional abbreviations

BRSUG, University of Bristol, School of Earth Sciences, Geology Collections, UK; EXEMS, Royal Albert Memorial Museum and Art Gallery, Exeter, UK; MCZ, Museum of Comparative Zoology, Harvard University, Cambridge, Massachusetts, USA.

### Scanning and segmentation

The three-dimensional (3D) skull and skeleton are based on two separate image stacks of differing resolution. First, a CT scan of BRSUG 29950-14 was produced using the Nikon XTH 225ST X-ray tomography scanner belonging to the Palaeobiology Research Group in the School of Earth Sciences, University of Bristol. Scanned in two parts at 224 kV, it produced a dataset containing a total of 2,940 slices at a voxel resolution of 25.59 μm.

Reconstruction of the skeleton was performed using Avizo (v.2020.2)[53] to generate 3D models. All surface models were generated in Avizo with an unconstrained smoothing factor of 2 and then exported in an ASCII standard triangle language format. These files were then imported into Blender (v.2.93.3)[54] for the purpose of taking photographs and model manipulation. Measurements were also taken in Blender, and the unit scaling was changed to 0.00109 to calibrate the model for accurate measurements of the 3D model.

Second, BRSUG 29950-14 was scanned using propagation phase-contrast synchrotron X-ray microcomputed tomography at the BM18 beamline of the European Synchrotron Radiation Facility (ESRF) in Grenoble, France. The following beamline set up was used: filtered white beam (Mo 3.75 mm); sample-detector distance of 6 m; indirect detector comprising a 2-mm-thick cerium-doped lutetium aluminium garnet scintillator ($Lu_3Al_5O_{12}$:Ce); ×0.642 magnification from a photographic lens; and a sCMOS IRIS-15 camera (Teledyne Photometrics). This set up resulted in a total integrated detected energy of 164 keV, a measured pixel size of 6.0 μm and a field of view of 5,056 × 1,744 (height × volume) pixels (that is, 30.34 × 10.46 mm). To compensate for the limited field of view, 17 acquisitions in planar circular modality with 360° rotation were performed, moving the specimen by 5 mm on the vertical axis after each turn, which resulted in a about 50% vertical overlap; the centre of rotation was shifted by 13.2 mm (corresponding to 2,200 pixels on the detector), which resulted in a reconstructed tomogram of 9,456 × 9,456 pixels (that is, 56.74 × 56.74 mm). Each acquisition consisted of 13,000 projections, each resulting from the accumulation of 5 frames with an exposure time of 25 ms (125 ms of total integration time per projection). Moreover, before each acquisition, 50 dark current images (image without beam) and 51 flatfield images (image with beam and no sample) were recorded. The tomographic reconstruction was done using the single distance phase retrieval approach[55,56] implemented in Nabu software[57]. Post-processing on the resulting 32 bits reconstructed stack included change of the dynamic range to 16 bits, discarding values outside the 0.001% minimum and 99.999% maximum percentiles, ring correction[58] and cropping of the volume. Moreover, a 2 × 2 × 2 binning copy of the data was generated to facilitate quick inspection. The MatLab code for ring correction, cropping and binning is available from GitHub (https://github.com/HiPCTProject/Tomo_Recon).

BRSUG 29950-14 was also scanned using conventional absorption-contrast tomography at the I12-JEEP beamline[59] (experiment MG40234-1) of the Diamond Light Source in Oxfordshire, UK. Imaging was performed using an 89 keV monochromatic beam, a sample-to-detector distance of 350 mm to minimize phase-contrast effects, optical module 2 (pixel size of 7.91 μm) with a 300 μm Crytur LuAG:Ce scintillator, a pco.edge 5.5 sCMOS visible light camera and an extended field of view configuration[60] (up to 40 × 12 mm for module 2). Each acquisition consisted of 4,801 projections over 360° (angular step of 0.075°) and an exposure time of 0.15 s per projection; a total of 4 acquisitions were taken with vertical offsets of 8 mm (around 33% overlap). Tomographic reconstruction was carried out using Savu software[61], with a pipeline involving flat and dark correction, correction for distortion introduced by the optical set-up, ring removal, automatic centre-of-rotation determination, sinogram stitching, Fresnel filtering and GPU reconstruction using the FBP algorithm implemented in Astra Toolbox[62]. Stitching of reconstructed volumes was performed using an overlap-informed digital image correlation technique implemented as part of an in-house script, with a final data volume of 40.5 × 40.5 × 36.3 mm (480 GB). Where necessary for ease of visualization, downsampling from float32 to uint8 was performed using an in-house script by fitting to a histogram of voxel intensities to a signal and noise model and clipping the resulting pixel intensity range to 5 standard deviations around the fitted data peak.

Reconstruction of the specimen's skull in 3D was carried out in the tomography laboratory housed in the University of Bristol Life Sciences Building using Dragonfly (v.2022.2)[63]. The size of the original image stack (15,088 slices) necessitated some image resampling for Dragonfly to process. Thus, the data were resampled at a factor of 4

in each axis to reduce the maximum number of slices to 3,771 while still retaining excellent detail of the skull and dentition. This higher resolution scan enabled the scoring of morphological characteristics that were not visible at lower resolution, in particular those of the teeth, vomer, palatine and pterygoid sequence.

## Phylogenetic analysis

Anatomical characteristics of BRSUG 29950-14 were scored against two data matrices for the purposes of this analysis.

First, the specimen was coded against a matrix of 383 morphological characteristics and 127 taxa (modified from a previous study[5]), including a new characteristic to code for the presence of a diapsid skull and 6 other taxa, including the procolophonids *K. bentoni*[51], *Procolophon trigoniceps*[64,65], *Palacrodon browni*[66] and the rhynchocephalians *C. hadroprodon*[30], *Wirtembergia hauboldae*[12] and *Parvosaurus harudensis*[35]. This matrix was chosen for its substantial outgroup taxa for early diapsids and archosaurs, the addition of new characteristics and taxa as well as the removal of others, building on previous versions of this same matrix[11,31,33]. *A. helsbypetrae* was scored for 203 of these characteristics and added to the matrix. Owing to the well-documented disagreement between morphology-only and molecular analyses[39,67–69], we adopted a strict topological constraint on groups in crown Squamata, using *S. punctatus* as the outgroup to follow the generally accepted topology resolved from recent molecular analyses. This constraint forces the position of Gekkonomorpha as basal to Lacertoidea with Iguania as the most derived clade of crown squamates, in line with recent publications[5,31]. Using this constraint enabled the resolution of the most accurate position of *A. helsbypetrae* in the larger topology of Lepidosauria as well as streamlining the analysis by bypassing the need for extensive processing time of molecular data.

Second, *A. helsbypetrae* was scored against a dataset comprising *Sophineta*, two extant squamates and 57 rhynchocephalians to assess its position in a more diverse rhynchocephalian phylogeny[29]. As in the original publication[29], several rogue taxa were excluded from subsequent analysis (*Whitakersaurus bermani*, *Deltadectes elvetica* and *Kawasphenodon expectatus*). Of the 162 morphological characteristics, 116 were successfully coded in *Agriodontosaurus* and added to the matrix. The matrix was analysed under equal-weights parsimony in TNT (v.1.6)[70] using a traditional heuristic search of 5,000 Wagner trees with tree bisection and reconnection as the branch swapping algorithm per a previous study[29]. The resulting most-parsimonious trees were exported to R (v.3.6.9)[71] to generate a 50% majority-rule consensus using the ape-package[72]. Further parsimony analyses were conducted using PAUP[73], with and without a genomic tree constraint.

## Bayesian inference

The modified 127 taxon matrix was subjected to Bayesian inference in MrBayes (v.3.2.7)[74] under the fossilized birth–death and relaxed clock transition model (outlined in a previous study[5]). The age calibration of individual taxa is based on a uniform distribution from their first and last appearance data[5,31]. Two runs of Markov chain Monte Carlo analysis were run for 30 million generations, sampled every 10,000 generations and a relative burn-in of 0.5. The resulting consensus tree was subject to further analysis in R[71], using the timePaleoPhy function of the paleotree package[75] to time calibrate it under the minimum branch length model and to produce a conservative estimate of branch divergence times, with a particular focus on the origin of Lepidosauromorpha. Dated time trees were constructed in R using the geoscalePhylo function of the strap package[76].

## Reporting summary

Further information on research design is available in the Nature Portfolio Reporting Summary linked to this article.

## Data availability

All data on specimen description and phylogenetic analyses are in Extended Data Figs. 1–9, Supplementary Information and Supplementary Data 1 and 2. The synchrotron X-ray CT data for the characterization of the skull of *A. helsbypetrae* BRSUG 29950-14 are available as raw acquisition data (https://doi.org/10.15151/ESRF-DC-2158672188)[77] and as processed data (https://doi.org/10.15151/ESRF-DC-2160804068)[78] from the European Synchrotron Radiation Facility website. Further data on phylogenetic data sets and analyses, as well as 3D segmented models from the CT scans are available at https://doi.org/10.5061/dryad.cvdncjth4 (ref. 79).

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

**Acknowledgements** We thank L. Martin-Silverstone for scanning the specimens at the XTM Facility, Palaeobiology Research Group, University of Bristol; staff at the ESRF for provision of synchrotron radiation facilities (beamline BM18), under proposal number ES1451, and the Diamond synchrotron under proposal number MG40234. M.J.B. was funded by the Natural Environment Research Council BETR programme (NE/P013724/1) European Research Council Advanced Grant 'Innovation' (ERC 788203).

**Author contributions** R.A.C. found the specimen and contributed information on occurrence. M.J.B. and D.I.W. designed the project and supervised two projects completed as part of the MSc in Palaeobiology at the University of Bristol, by T.S. (partial project) and D.M. (entire project). T.S. segmented the micro-CT scan made in Bristol, and D.M. segmented the synchrotron scan from the ESRF. V.F. and E.N. carried out the synchrotron scan at the ESRF in Grenoble, and A.L. carried out the synchrotron scan at Diamond, Harwell. D.M., D.I.W. and M.J.B. led the writing of the manuscript, and D.M. prepared the figures. All authors contributed to checking and revision of the manuscript.

**Competing interests** The authors declare no competing interests.

**Additional information**
**Correspondence and requests for materials** should be addressed to Daniel Marke or Michael J. Benton.

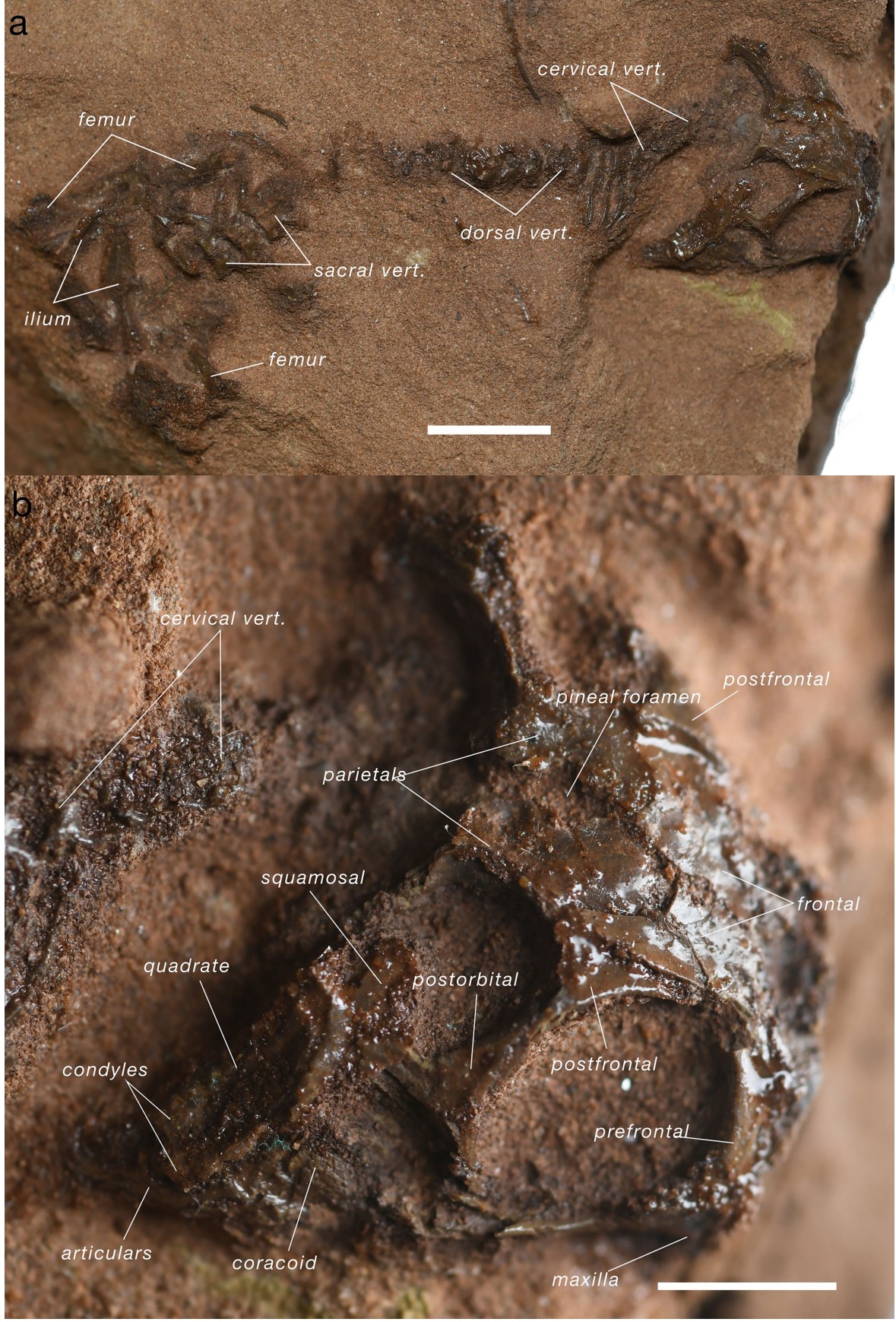

**Extended Data Fig. 1 | Holotype specimen of *Agriodontosaurus helsbypetrae*, BRSUG 29950-14. a**, overview of complete specimen, and **b**, close-up photograph of the preserved skull of *A. helsbypetrae*. Abbreviations: vert, vertebrae. Scale bars: **a**, 10 mm; **b**, 5 mm.

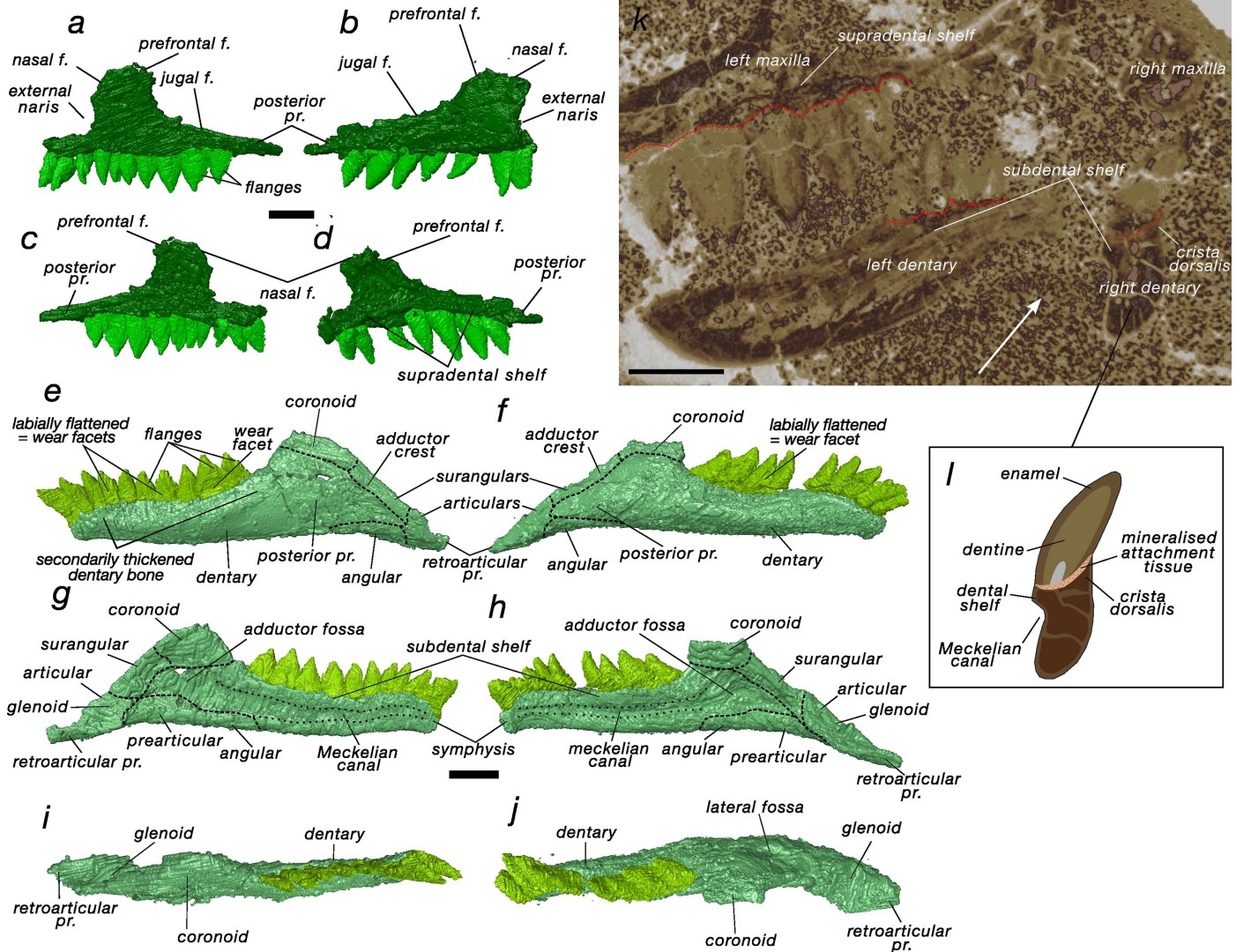

**Extended Data Fig. 2 | Three-dimensional reconstructions of the tooth bearing bones of *Agriodontosaurus helsbyetrae*.** Left (**a**, **c**) and right (**b**, **d**) maxillae in lateral and medial views, respectively. Left (**e**, **g**, **i**) and right (**f**, **h**, **j**) mandibles in lateral, medial and dorsal views, respectively. **k**, cross-section of tooth bearing bones of *A. helsbypetrae*; and **l**, illustration of right dentary cross-section. Abbreviations: f, facet; pr, process. Resolution is 6 μm voxel. Scale bars: 2 mm.

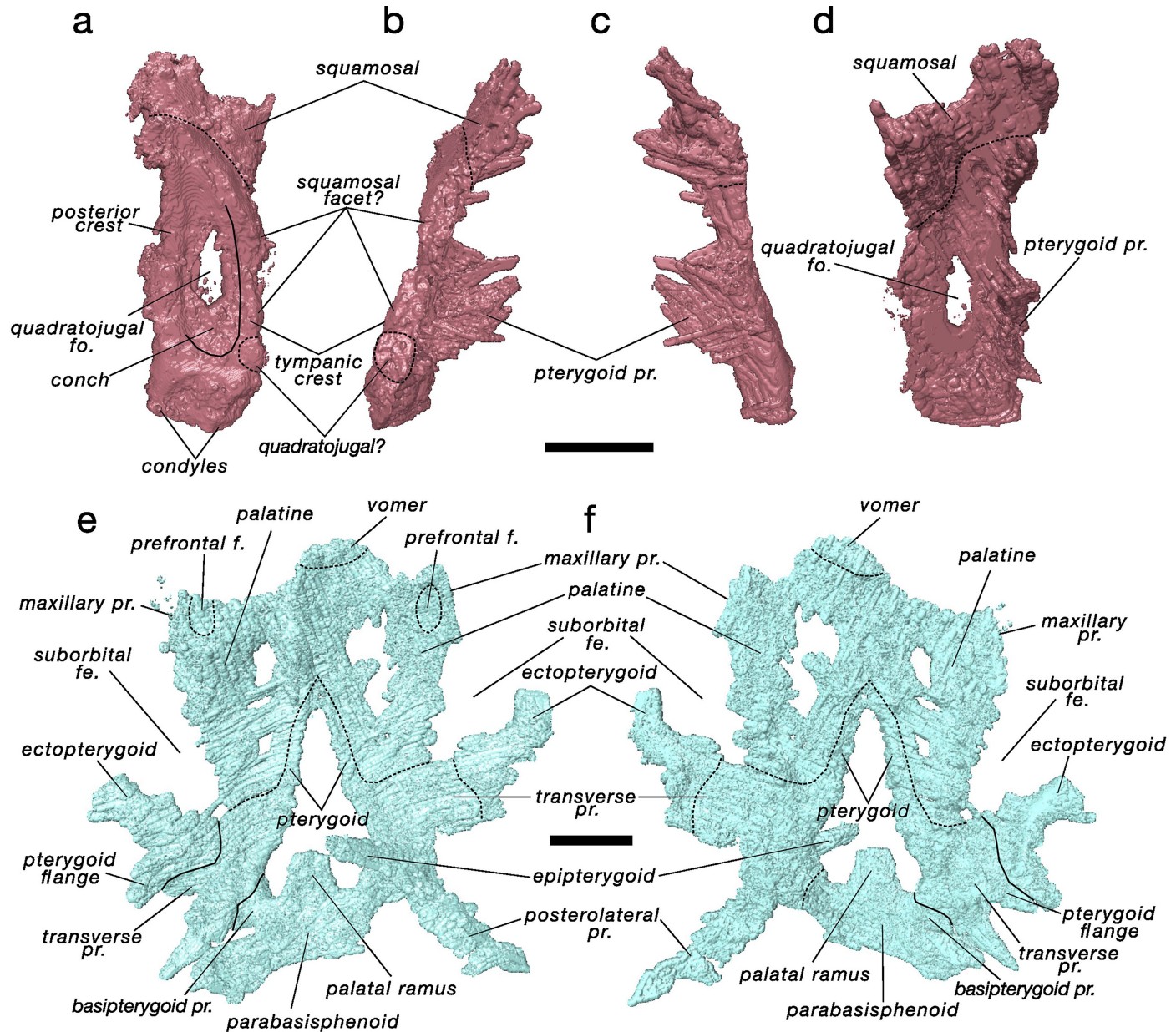

**Extended Data Fig. 3 | 3D reconstruction of the right quadrate and palate of *Agriodontosaurus helsbypetrae*.** Right quadrate in **a**, lateral; **b**, anterior; **c**, posterior; and **d**, medial views and the palatal sequence in **e**, dorsal; and **f**, ventral views. Solid lines represent observable contacts between bones, dashed lines are speculative contacts. Abbreviations: f, facet; fe, fenestra; fo, foramen; pr, process. Resolution is 6 μm voxel. Scale bars: 2 mm.

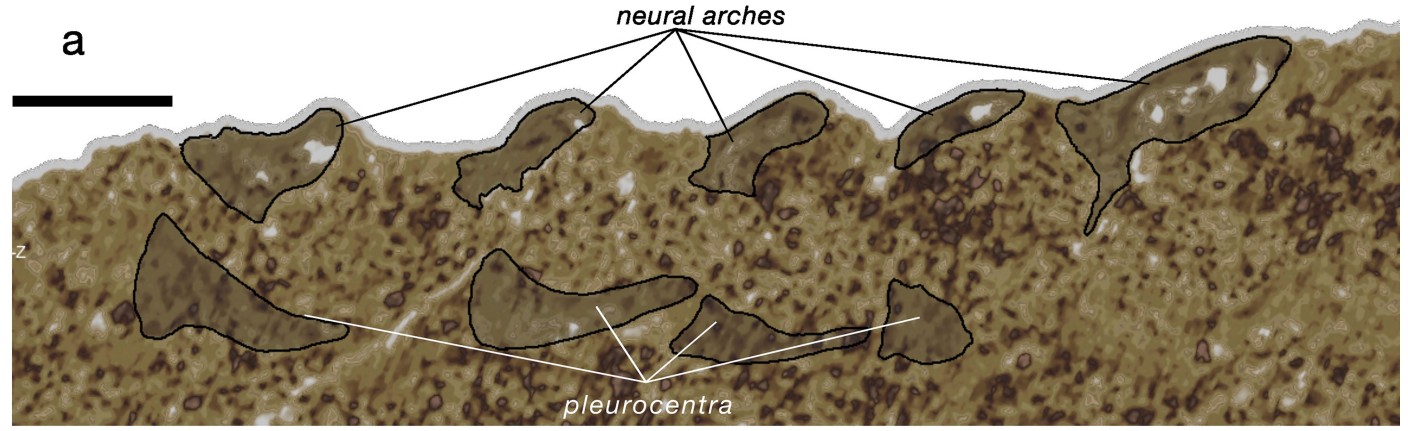

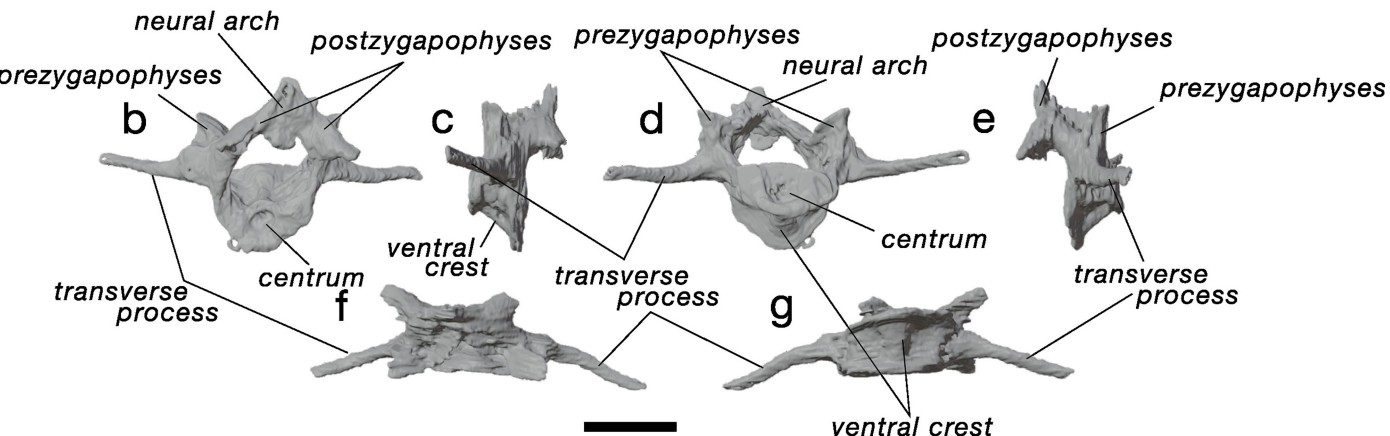

**Extended Data Fig. 4 | Lateral cross-section of articulated cervical vertebrae (a) and 3D reconstruction of isolated presacral vertebrae (b-g) of *Agriodontosaurus helsbypetrae*.** Cross-section shows the amphicoelic and notochordal vertebrae. 3D images in **b**, anterior; **c**, right lateral; **d**, posterior; **e**, left lateral; **f**, dorsal and **g**, ventral views. Resolution is 26 µm voxel. Scale bar: 2 mm.

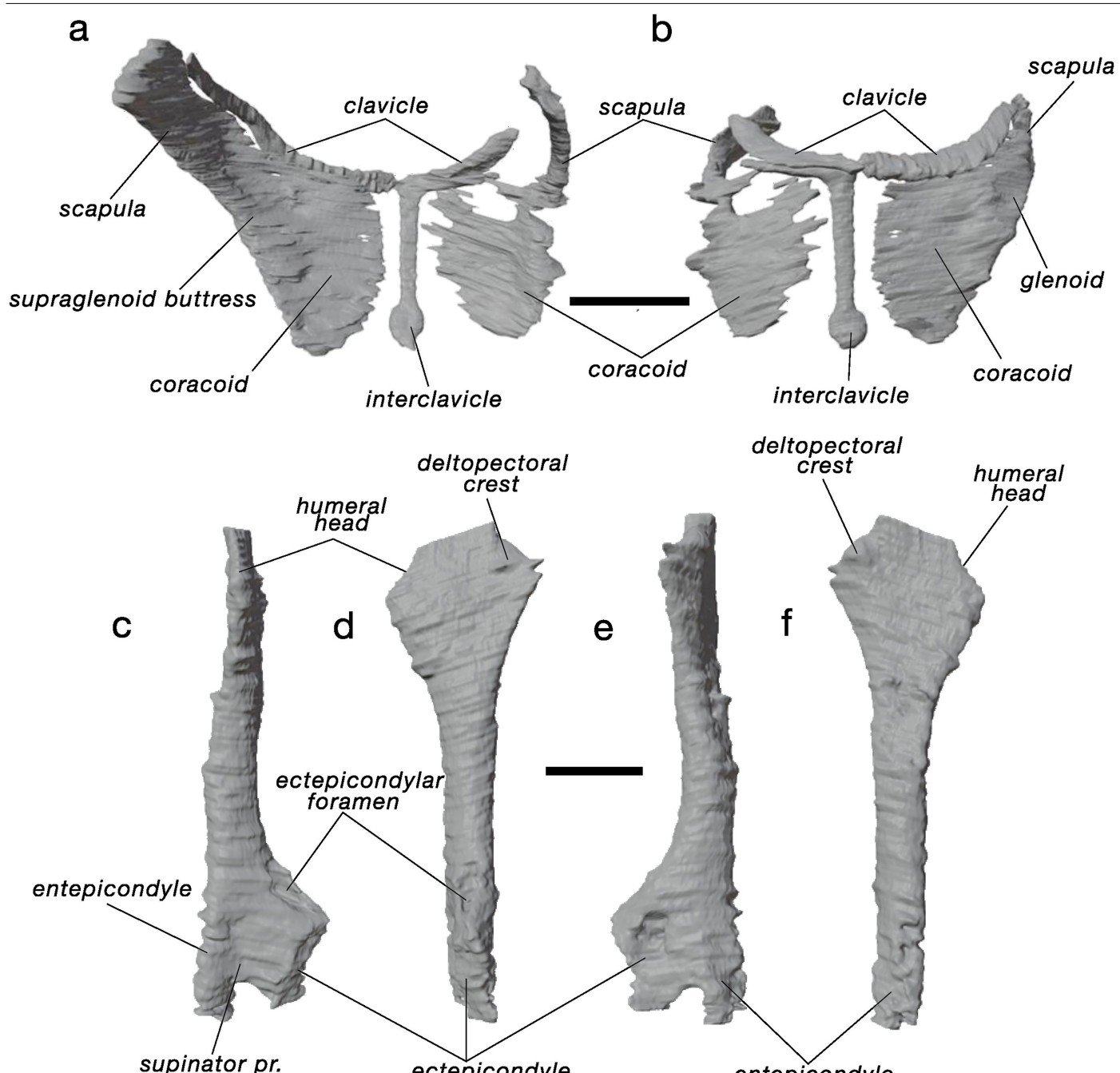

**Extended Data Fig. 5 | 3D reconstruction of the pectoral girdle (a-b) and left humerus (c-f) of *Agriodontosaurus helsbypetrae*.** Articulated interclavicle, clavicles, scapulae and coracoids in **a**, dorsal; and **b**, ventral views. Humerus shown in **c**, lateral; **d**, anterior; **e**, medial and **f**, posterior views. Abbreviations: pr., process. Resolution is 26 µm voxel. Scale bars: 2 mm. Resolution is 26 µm voxel. Scale bars: 2 mm.

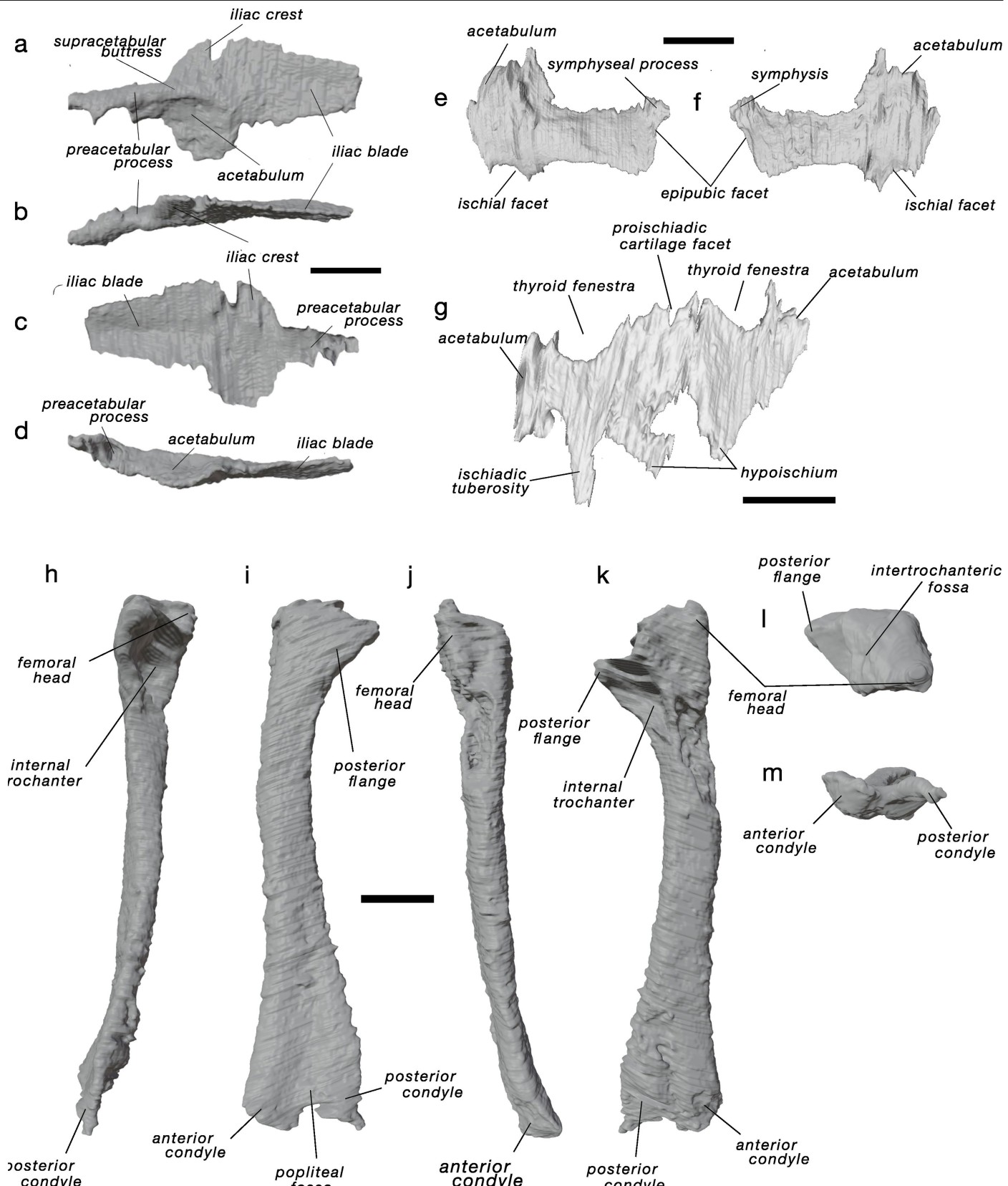

**Extended Data Fig. 6 | 3D reconstruction of isolated elements of the pelvic girdle (a–g) and left femur (h–m) of *Agriodontosaurus helsbypetrae*.** Left ilium in **a**, lateral; **b**, dorsal; **c**, medial and **d**, ventral views. Right pubis in **e**, lateral; and **f**, medial views. Ischia in **g**, dorsal view. Left femur in **h**, posterior; **i**, lateral; **j**, anterior; **k**, medial; **l**, dorsal and **m**, ventral views. Resolution is 26 μm voxel. Scale bars: 2 mm.

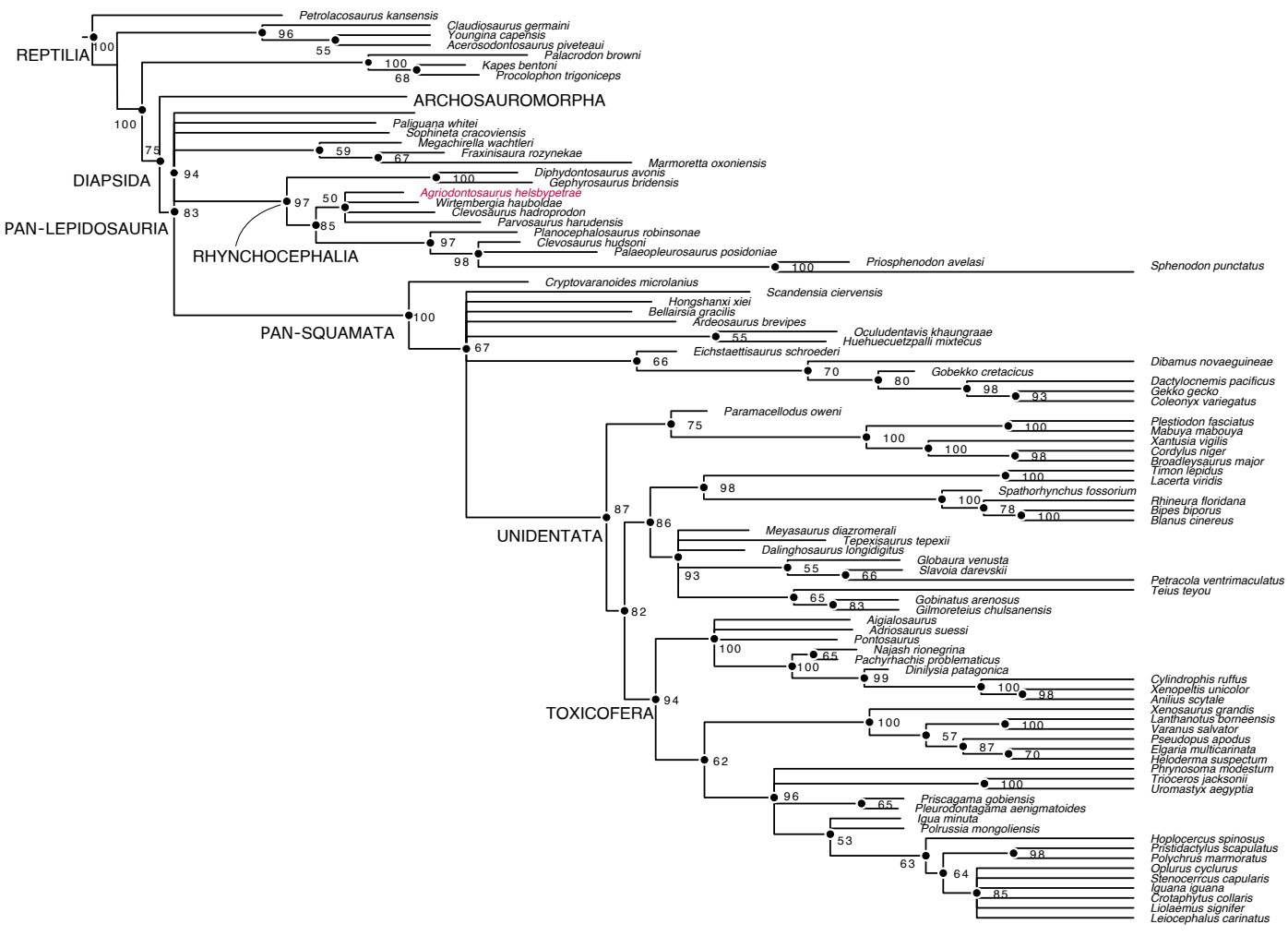

**Extended Data Fig. 7 | Bayesian phylogenetic tree.** Extended phylogenetic tree from Bayesian analysis of the modified Tałanda et al.[5] diapsid dataset, resolving *Agriodontosaurus helsbypetrae* within Sphenodontia. Numbers at nodes represent node posterior probability (percentage proportions).

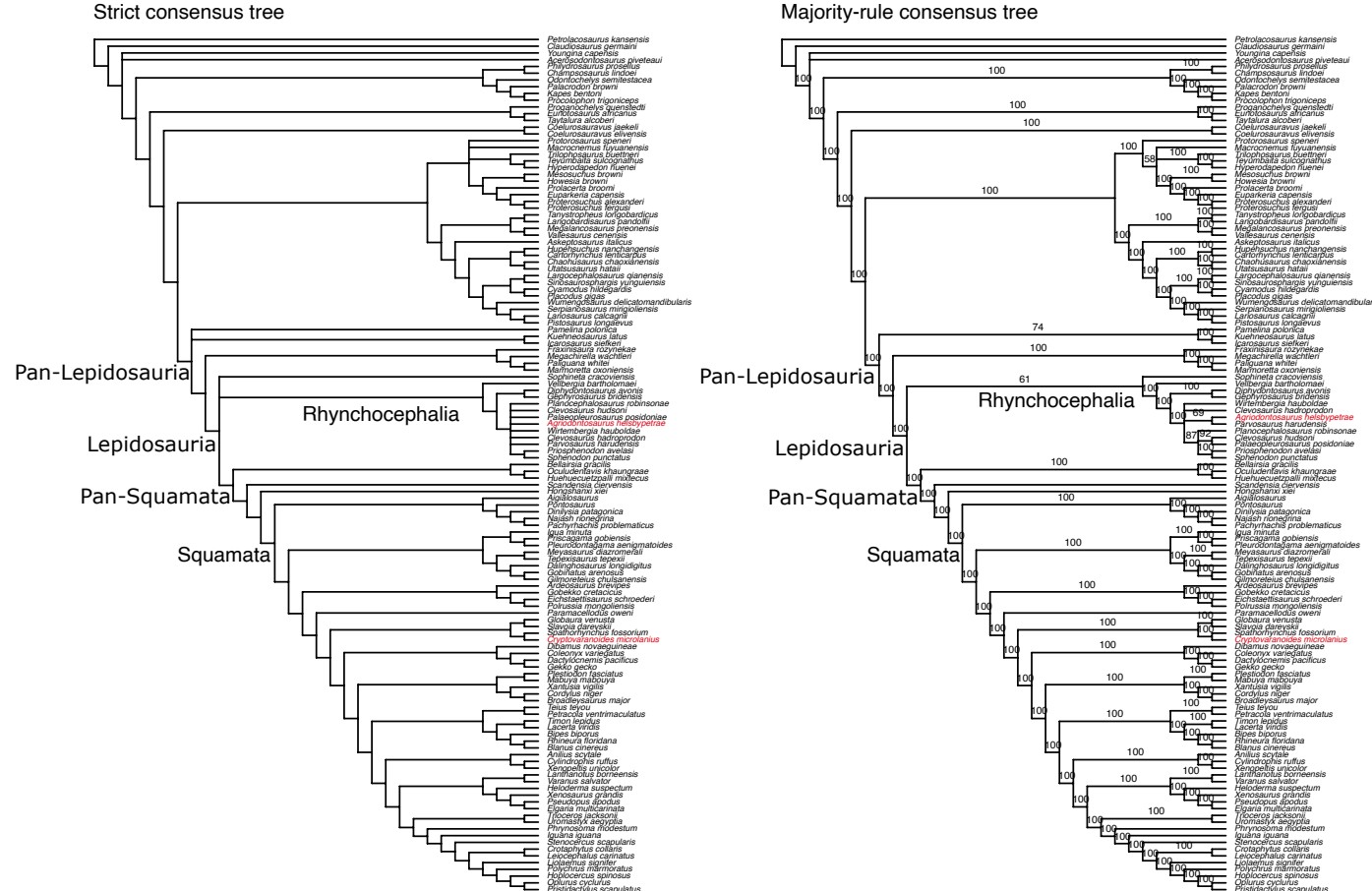

**Extended Data Fig. 8 | Parsimony-based phylogenetic trees.** Strict consensus and 50% majority trees showing the position of *Agriodontosaurus helsbypetrae* and *Cryptovaranoides microlanius* recovered from parsimony analysis in PAUP using the Talanda et al. (2022) dataset of diapsids; trees using constraint of molecular-based phylogeny. Numbers at nodes represent clade frequency.

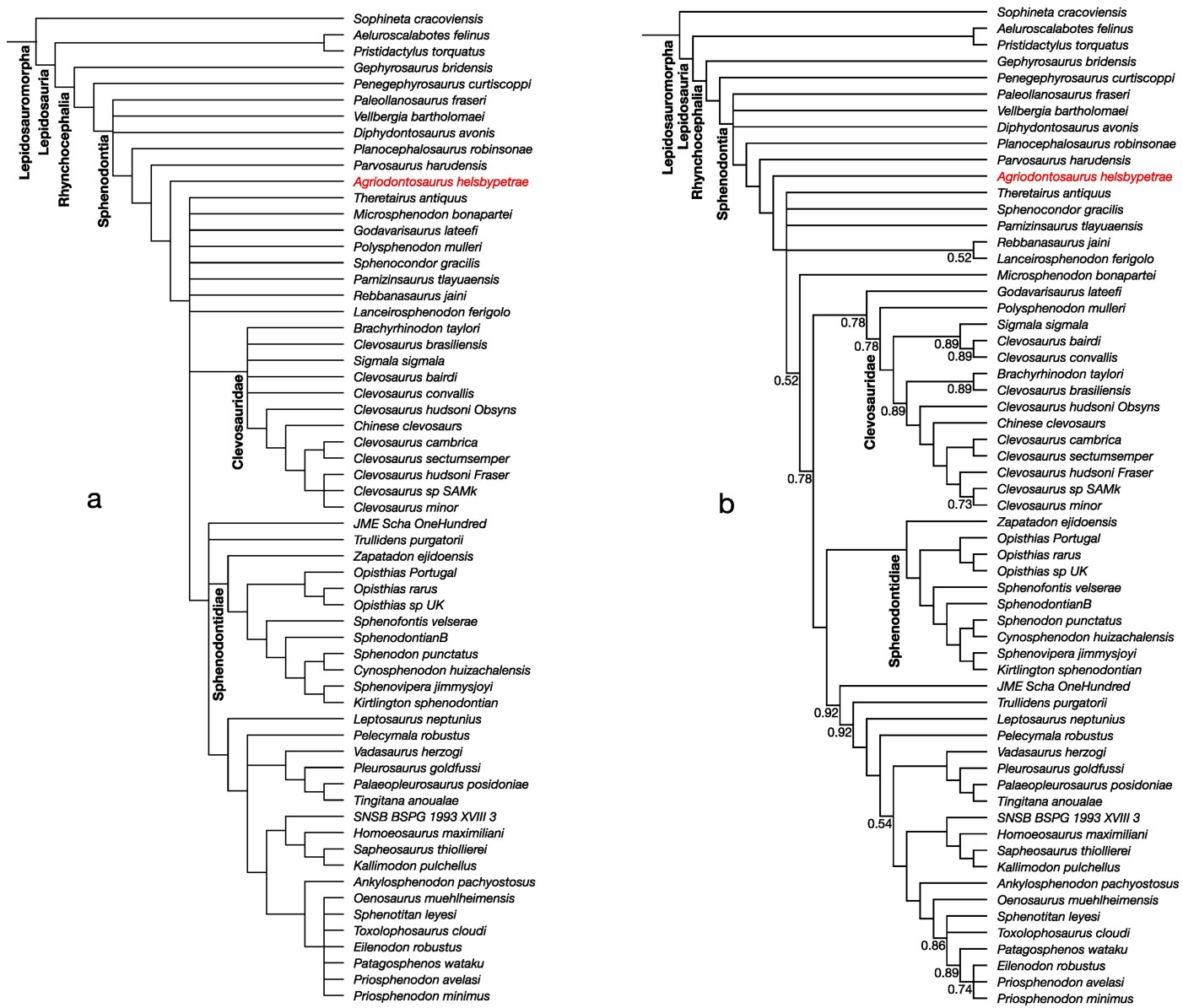

**Extended Data Fig. 9 | Parsimony-based phylogenetic trees. a**, strict consensus and **b**, 50% majority trees showing the position of *Agriodontosaurus helsbypetrae* recovered from parsimony analysis in TNT using the Chambi-Trowell (2021) dataset of rhynchocephalians. Numbers at nodes represent clade frequency.

# Reporting Summary

## Statistics

For all statistical analyses, confirm that the following items are present in the figure legend, table legend, main text, or Methods section.

| n/a | Confirmed | |
|---|---|---|
| ☐ | ☒ | The exact sample size (*n*) for each experimental group/condition, given as a discrete number and unit of measurement |
| ☒ | ☐ | A statement on whether measurements were taken from distinct samples or whether the same sample was measured repeatedly |
| ☒ | ☐ | The statistical test(s) used AND whether they are one- or two-sided *Only common tests should be described solely by name; describe more complex techniques in the Methods section.* |
| ☒ | ☐ | A description of all covariates tested |
| ☒ | ☐ | A description of any assumptions or corrections, such as tests of normality and adjustment for multiple comparisons |
| ☒ | ☐ | A full description of the statistical parameters including central tendency (e.g. means) or other basic estimates (e.g. regression coefficient) AND variation (e.g. standard deviation) or associated estimates of uncertainty (e.g. confidence intervals) |
| ☒ | ☐ | For null hypothesis testing, the test statistic (e.g. *F*, *t*, *r*) with confidence intervals, effect sizes, degrees of freedom and *P* value noted *Give P values as exact values whenever suitable.* |
| ☐ | ☒ | For Bayesian analysis, information on the choice of priors and Markov chain Monte Carlo settings |
| ☒ | ☐ | For hierarchical and complex designs, identification of the appropriate level for tests and full reporting of outcomes |
| ☒ | ☐ | Estimates of effect sizes (e.g. Cohen's *d*, Pearson's *r*), indicating how they were calculated |

*Our web collection on statistics for biologists contains articles on many of the points above.*

## Software and code

Policy information about availability of computer code

| Data collection | Dragonfly v. 2022.2; Avizo 2020.2 |
|---|---|
| Data analysis | Standard phylogenetic analysis tools (PAUP, tnt v.1.6, MrBayes v.3.2.7) and programs in R (ape, paleotree). |

For manuscripts utilizing custom algorithms or software that are central to the research but not yet described in published literature, software must be made available to editors and reviewers. We strongly encourage code deposition in a community repository (e.g. GitHub). See the Nature Portfolio guidelines for submitting code & software for further information.

## Data

Policy information about availability of data

All manuscripts must include a data availability statement. This statement should provide the following information, where applicable:
- Accession codes, unique identifiers, or web links for publicly available datasets
- A description of any restrictions on data availability
- For clinical datasets or third party data, please ensure that the statement adheres to our policy

All data on specimen description and phylogenetic analysis are in the Extended data and SI. In addition, synchrotron X-ray CT data for the characterisation of the skull of Agriodontosaurus helsbypetrae BRSUG 29950-14 are available as raw acquisition data (doi.org/10.15151/esrf-dc-2158672188) and as processed data (doi.esrf.fr/10.15151/ESRF-DC-2160804068). Please cite these as:
Benton, M., & Fernandez, V. (2025). Synchrotron X-ray CT raw data for the characterization of the skull of Agriodontosaurus helsbypetrae BRSUG 29950-14. (Version

1) [Dataset]. European Synchrotron Radiation Facility. doi.org/10.15151/ESRF-DC-2158672188

Benton, M. J., & Fernandez, V. (2025). Synchrotron X-ray CT processed data of the skull of Agriodontosaurus helsbypetrae BRSUG 29950-14. (Version 1) [Dataset]. European Synchrotron Radiation Facility. doi.org/10.15151/ESRF-DC-2160804068

All data on specimen description and phylogenetic analysis are in the Extended data and SI. In addition, synchrotron X-ray CT data for the characterisation of the skull of Agriodontosaurus helsbypetrae BRSUG 29950-14 are available as raw acquisition data (doi.org/10.15151/esrf-dc-2158672188) and as processed data (doi.esrf.fr/10.15151/ESRF-DC-2160804068). Please cite these as:

Benton, M., & Fernandez, V. (2025). Synchrotron X-ray CT raw data for the characterization of the skull of Agriodontosaurus helsbypetrae BRSUG 29950-14. (Version 1) [Dataset]. European Synchrotron Radiation Facility. doi.org/10.15151/ESRF-DC-2158672188

Benton, M. J., & Fernandez, V. (2025). Synchrotron X-ray CT processed data of the skull of Agriodontosaurus helsbypetrae BRSUG 29950-14. (Version 1) [Dataset]. European Synchrotron Radiation Facility. doi.org/10.15151/ESRF-DC-2160804068

## Research involving human participants, their data, or biological material

Policy information about studies with [human participants or human data](). See also policy information about [sex, gender (identity/presentation), and sexual orientation]() and [race, ethnicity and racism]().

| Reporting on sex and gender | N/a |
|---|---|
| Reporting on race, ethnicity, or other socially relevant groupings | N/a |
| Population characteristics | N/a |
| Recruitment | N/a |
| Ethics oversight | N/a |

Note that full information on the approval of the study protocol must also be provided in the manuscript.

# Field-specific reporting

Please select the one below that is the best fit for your research. If you are not sure, read the appropriate sections before making your selection.

☐ Life sciences  ☐ Behavioural & social sciences  ☒ Ecological, evolutionary & environmental sciences

For a reference copy of the document with all sections, see [nature.com/documents/nr-reporting-summary-flat.pdf]()

# Ecological, evolutionary & environmental sciences study design

All studies must disclose on these points even when the disclosure is negative.

| Study description | Description of fossil reptile; phylogenetic analysis of morphological traits |
|---|---|
| Research sample | One specimen |
| Sampling strategy | n/a |
| Data collection | Cladistic data matrix taken from Talanda et al. (2022) and modified as described in the Methods. |
| Timing and spatial scale | Phylogenetic data spans from Carboniferous to Recent, and extends spatially worldwide |
| Data exclusions | None |
| Reproducibility | We provide the data matrix and nexus codes for analysis |
| Randomization | n/a |
| Blinding | n/a |

Did the study involve field work?  ☐ Yes  ☒ No

# Reporting for specific materials, systems and methods

We require information from authors about some types of materials, experimental systems and methods used in many studies. Here, indicate whether each material, system or method listed is relevant to your study. If you are not sure if a list item applies to your research, read the appropriate section before selecting a response.

## Materials & experimental systems

| n/a | Involved in the study |
|-----|----------------------|
| ☒ | ☐ Antibodies |
| ☒ | ☐ Eukaryotic cell lines |
| ☐ | ☒ Palaeontology and archaeology |
| ☒ | ☐ Animals and other organisms |
| ☒ | ☐ Clinical data |
| ☒ | ☐ Dual use research of concern |
| ☒ | ☐ Plants |

## Methods

| n/a | Involved in the study |
|-----|----------------------|
| ☒ | ☐ ChIP-seq |
| ☒ | ☐ Flow cytometry |
| ☒ | ☐ MRI-based neuroimaging |

# Palaeontology and Archaeology

| | |
|---|---|
| Specimen provenance | Specimen comes from the Triassic on the Devon coast, on public land |
| Specimen deposition | Specimen is lodged in University of Bristol geological collections, repository code BRSUG |
| Dating methods | Published biostratigraphy and magnetostratigraphy data; we did no dating |

☐ Tick this box to confirm that the raw and calibrated dates are available in the paper or in Supplementary Information.

| | |
|---|---|
| Ethics oversight | University of Bristol |

Note that full information on the approval of the study protocol must also be provided in the manuscript.

# Plants

| | |
|---|---|
| Seed stocks | n/a |
| Novel plant genotypes | n/a |
| Authentication | n/a |

