## [Peer Review File · Nature]

The oldest lepidosaur and origins of lepidosaur feeding adaptations

Corresponding Author: Professor Michael Benton

Version 0:

Reviewer comments:

Referee #1

(Remarks to the Author)

This manuscript describes a new fossil lepidosaur from the Triassic of the southern UK. Originating from the Anisian, this would be the oldest lepidosaur known to date and could therefore play a key role in elucidating the early evolutionary history of lepidosaurs.

The evidence presented to support *Agriodontosaurus* as a Rhynchocephalian is quite compelling. Certainly the fossil is very intriguing and the dentition seems to be very unusual and rather surprising in a tetrapod that is so small.

I have two main points to raise:

Firstly the authors need to clarify the size of the specimen. On l. 85 they say the skull is estimated to be 14 mm long, yet based on the scale bar in Figure 3 the complete skull would be more like 25+mm long. While still a tiny animal, this is a significant difference for the authors claim of a diet of large insects. For the most part "large" insects from the Triassic include some blattoids, orthopterans and hemipterans which might be up to 15mm long – quite a mouthful for an animal with a 14 mm long skull and a pretty rigid skull and difficult to really imagine. Even at 25 mm long it would be quite a task. However, the authors actually take it a step further on l. 410 and 411, and suggest it may have tackled insects with a 150mm wing length. That does seem to be an extraordinary claim! Tackling a large prey item that was its equal in size is difficult to conceive - even if scavenging could an animal with a 14 mm mandible bite into and break into chunks such a large keratinised body? I really doubt it.

The second point concerns the nature of the palate. In the analyses there appears to be no evidence for a palatal dentition and this lies at the centre of their discussions. However, I would like to understand how well preserved the specimen is – it does seem possible that the palatine bones are incomplete. The reconstruction of the palatines and ectopterygoid (Figure 3i) seem to be more definitive than the 3 D model (Figure 3b) would suggest. What evidence do the authors have that what is preserved of the skull is in good condition with finished bone surfaces. An examination of the image of the block that is included in the supplementary data indicates some areas of the skeleton are more fragmentary than others. Indeed on l. 159, 160 the authors even state that "the sutures between elements are obscure and cannot be discerned.....partially because of skull deformation ."

As a result, I don't necessarily think the authors have provided strong evidence to support their claim for a lack of palatal teeth, including an enlarged row of teeth on the palatine. Absence of evidence is not evidence of absence.

More minor comments

Abstract l. 15 refers to the Otter Sandstone. Why not Helsby Sandstone Formation, as used in l. 79

l. 31 does the gap between the jugal and quadrate allow the muscles to bulge. Don't the temporal fenestrae play a more significant role?

L 245. In what way are the teeth flanged? There are very clear descriptions of flange size and orientation of flanges on mandibular and maxillary teeth in many other rhynchocephalians. Does the pattern match any of these? The remarkable size of the teeth requires a little more detail in the description. Are these slicing and what is the distribution of wear facets on these teeth? This is not clearly shown.

l. 307 Delete "of"

l. 347 states that the skull is less robust in the postorbital region than *Gephyrosaurus* and *Diphyodontosaurus*. Doesn't this therefore speak against a powerful bite?

Referee #2

(Remarks to the Author)

This is a really interesting discovery that confirms previous suggestions that lepidosaurs arose in the Early or early Middle Triassic. The new taxon has some intriguing features such as the mode of tooth implantation.

Generally, the paper is excellent with a nice, concise description of the material. I have made a few comments on the review copy, mostly concerning proper anatomical terminology.

Figure 3 is much too dark and might be unintelligible in the reduced printed version. Subfigure 3e has several points that should be fixed: The teeth do not resemble those in the scan images and even those that are not en echelon are tightly packed together. The squamosal is rendered too small and the quadrate-quadratojugal is incorrectly oriented, both creating an odd-looking suspensorial region. You might want to consult reconstructions of the skulls of other early-diverging rhynchocephalians such as *Diphydontosaurus* (Whiteside, 1986).

Comments on text

l. 16 'oldest currently known', not 'current oldest'

l. 18 delete 'giant' – the small size of the skull would not allow capturing a titanopteran

l. 151 'quadratojugal foramen' is the generally used term (e.g., Whiteside, 1986)

l. 194 Tetrapods have only one mandible, which comprises two 'mandibular rami' or hemimandibles. Only arthropods have 'mandibles'.

l. 226 There is no such thing as 'lateral mandible'. Do you mean 'lateral surface of the mandibular ramus'?

l. 255-256 The mode of a tooth implantation is very different from other early rhynchocephalians such as *Diphydontosaurus* and *Wirtembergia*, in which the anterior teeth have pleurodont tooth implantation and the posterior ones have acrodont attachment.

l. 347 Do you mean 'less robust posterior portion of the skull'?

l. 414 There are clear scouring marks on some jaws of *Wirtembergia* – they do not 'appear' to be there.

l.431-432 This sentence is unsupported by a palaeobiogeographical analysis and, in any case, meaningless as Europe did not exist as a geographical entity during the Triassic Period. Please delete this sentence.

Figure 3 is much too dark and will likely be unintelligible in the reduced printed version. 3e has several points that should be fixed: The teeth do not resemble those in the scan images and even those that are not en echelon are too tightly packed together. The squamosal is rendered too small and the quadrate-quadratojugal is incorrectly oriented, both creating an odd-looking suspensorial region. You might want to consult reconstructions of the skulls of other early-diverging rhynchocephalians such as *Diphydontosaurus* (Whiteside, 1986).

Version 1:

Reviewer comments:

Referee #1

(Remarks to the Author)

Many thanks for the opportunity to review the revised version of this very interesting manuscript. I am pleased to see that the authors have made every effort to address my initial suggestions and concerns.

I still question the absence of palatal teeth, but I thank the authors for providing further clarification on their original reasoning. I accept they have made as good a case as they can with the provision of new scans and some clarifications around the absence of tooth wear facets. This is very helpful and it does strengthen their case. While I still consider the presence or absence of palatal teeth to be a key part of the manuscript, the support the authors have provided for their point of view is reasonable.

Referee #2

(Remarks to the Author)

Thank you for fully addressing the comments from my previous review.

Dear Mike

Your manuscript, "The oldest lepidosaur and origins of lepidosaur feeding adaptations", has now been seen by two referees, whose comments are attached below. While they find your work of potential interest, as do we, they have raised important concerns that in our view need to be addressed before we can consider publication in Nature.

We work through these concerns and provide detailed responses to each, below.

Should further work allow you to address these criticisms, we would be happy to consider a revised manuscript (unless something similar has been accepted at Nature or appeared elsewhere in the meantime). We hope to receive your revised paper within four to six months. If you cannot complete the required revisions within this time frame, please let us know when you would anticipate being able to submit a revised manuscript.

In the meantime, we hope that you will find our referees' comments helpful. Please do not hesitate to contact me if there is anything you would like to discuss.

Yours sincerely

Henry

Dr Henry Gee

Senior Editor Biological Sciences

nature@nature.com

<http://www.nature.com/nature>

See Nature's new guide to authors at www.nature.com/nature/submit

Referees' comments:

Referee #1 (Remarks to the Author):

This manuscript describes a new fossil lepidosaur from the Triassic of the southern UK. Originating from the Anisian, this would be the oldest lepidosaur known to date and could therefore play a key role in elucidating the early evolutionary history of lepidosaurs.

The evidence presented to support Agriodontosaurus as a Rhynchocephalian is quite compelling. Certainly the fossil is very intriguing and the dentition seems to be very unusual and rather surprising in a tetrapod that is so small.

Many thanks for the positive remarks

I have two main points to raise:

Firstly the authors need to clarify the size of the specimen. On l. 85 they say the skull is estimated to be 14 mm long, yet based on the scale bar in Figure 3 the complete skull would be more like 25+mm long. While still a tiny animal, this is a significant difference for the authors claim of a diet of large insects. For the most part "large" insects from the Triassic include some blattoids, orthopterans and hemipterans which might be up to 15mm long – quite a mouthful for an animal with a 14 mm long skull and a pretty rigid skull and difficult to really imagine. Even at 25 mm long it would be quite a task. However, the authors actually take it a step further on l. 410 and 411, and suggest it may have tackled insects with a 150mm wing length. That does seem to be an extraordinary claim! Tackling a large prey item that was its equal in size is difficult to conceive - even if scavenging could an animal with a 14 mm mandible bite into and break into chunks such a large keratinised body? I really doubt it.

The 10 mm scale bar noted in the caption of Figure 3 was in error and has been corrected to 5 mm. Apologies for this error. We have also checked all other scale bars and size indications to ensure there are no other such bloopers.

We have omitted the comment on lines 410-412 regarding the largest Mesozoic insects to avoid the interpretation described by the reviewer. It is indeed extremely unlikely that the **largest** representatives of these groups were the prey items of choice for *Agriodontosaurus* and this was not the intended suggestion. Thank you for raising this.

In terms of dealing with smaller prey up to 15 mm as mentioned by the reviewer, we suggest in l. 404 that this animal was a lingual feeder like *Sphenodon* which itself has an immobile skull and that a mobile skull was less important for prey prehension in *Agriodontosaurus*.

The second point concerns the nature of the palate. In the analyses there appears to be no evidence for a palatal dentition and this lies at the centre of their discussions. However, I would like to understand how well preserved the specimen is – it does seem possible that the palatine bones are incomplete. The reconstruction of the palatines and ectopterygoid (Figure 3i) seem to be more definitive than the 3D model (Figure 3b) would suggest. What evidence do the authors have that what is preserved of the skull is in good condition with finished bone surfaces. An examination of the image of the block that is included in the supplementary data indicates some areas of the skeleton are more fragmentary than others. Indeed on l. 159, 160 the authors even state that “the sutures between elements are obscure and cannot be discerned... partially because of skull deformation .”

As a result, I don't necessarily think the authors have provided strong evidence to support their claim for a lack of palatal teeth, including an enlarged row of teeth on the palatine. Absence of evidence is not evidence of absence.

As a result of the reviews, we looked at the slices across the palate to include the region lying medially of the maxilla teeth at the point of occlusion with the dentary teeth (images now included in Supplements 2, 3). We can clearly recognise the large maxillary teeth and the occluding dentary teeth on the right-hand side of the skull. Bones of the palate, including the palatine and the pterygoid are recognisable but there are no large teeth on the palatine or pterygoid. This corresponds with our recognition that there are no wear facets on the lingual sides of the dentary teeth. Lingual tooth wear facets are found on the dentary in rhynchocephalians that have a lateral tooth row on the palatine, e.g. in clevosaurs.

To further address the reviewer's concerns, we have produced a series of images (PDF files) through the palate (in dorso-ventral and in coronal slices) to demonstrate our conclusion that *Agriodontosaurus* lacks large palatal teeth. These include annotated images through the ESRF scan that was used initially as well as a brand-new image stack acquired from Diamond Light Source Ltd, Oxfordshire, for the purposes of this response. We also include an MP4 file of the image stack for the benefit of the reviewers, to be included in conjunction with the extended data and the annotated slices.

We are aware that 'absence of evidence does not mean evidence of absence', and we cannot therefore rule out that *Agriodontosaurus* had palatal teeth. However, we can indicate two lines of evidence: (1) absence of lingual tooth wear facets from putative palatal teeth on dentary teeth; and (2) re-investigation now from two micro-CT studies, one from ESRF and a new one from Diamond, where we identify the palatine and pterygoid bones, but find no hint of teeth.

We reword and expand this section.

More minor comments

Abstract l. 15 refers to the Otter Sandstone. Why not Helsby Sandstone Formation, as used in l. 79

Amended

l. 31 does the gap between the jugal and quadrate allow the muscles to bulge. Don't the temporal fenestrae play a more significant role?

We removed this remark; it's enough to say the open lower temporal bar is essential for cranial kinesis.

L 245. In what way are the teeth flanged? There are very clear descriptions of flange size and orientation of flanges on mandibular and maxillary teeth in many other rhynchocephalians. Does the pattern match any of these? The remarkable size of the teeth requires a little more detail in the description. Are these slicing and what is the distribution of wear facets on these teeth? This is not clearly shown.

We have provided more detail on the flanges of the posterior teeth of *Agriodontosaurus*. We now make clear that the maxillary and dentary posterior teeth each have a postero-lingual flange but lack mesial flanges (Fig. 3F–G; Extended Data Fig. 2E–H) which are present in clevosaurids such as *C. hudsoni*. We also reference the wear facets found on the labial side of the dentary that occluded with the large, flanged maxillary teeth.

I. 307 Delete “of”
corrected to “of the pterygoid”

I. 347 states that the skull is less robust in the postorbital region than *Gephyrosaurus* and *Diphydontosaurus*. Doesn't this therefore speak against a powerful bite?

We have improved the reconstruction of *Agriodontosaurus* reducing the area (in lateral view) of the lower temporal fenestra and made the squamosal more robust, although it is somewhat less robust than in *Diphydontosaurus*. There are strong wear facets on the dentary teeth which we have annotated in the Extended data Figure 2 and such wear facets will be caused by hard biting of dentary against maxillary teeth. We show the precise occlusion between the dentary and maxillary teeth in the CT slices of the additional data.

Referee #2 (Remarks to the Author):

This is a really interesting discovery that confirms previous suggestions that lepidosaurs arose in the Early or early Middle Triassic. The new taxon has some intriguing features such as the mode of tooth implantation.

Generally, the paper is excellent with a nice, concise description of the material. I have made a few comments on the review copy, mostly concerning proper anatomical terminology.

Many thanks for these overview comments.

Figure 3 is much too dark and might be unintelligible in the reduced printed version. Subfigure 3e has several points that should be fixed: The teeth do not resemble those in the scan images and even those that are not in an echelon are tightly packed together. The squamosal is rendered too small and the quadrate-quadratojugal is incorrectly oriented, both creating an odd-looking suspensorial region. You might want to consult reconstructions of the skulls of other early-diverging rhynchocephalians such as *Diphydontosaurus* (Whiteside, 1986).

We thank the reviewer for raising these points. We have lightened Fig. 3. This has been addressed by further information from CT slices and scrutiny of the scans, and we have reorientated the quadrate suspensorium so that it accords with the reconstructions of basal rhynchocephalians such as *Diphydontosaurus* and *Gephyrosaurus*.

We have superimposed our reconstruction of the maxillary teeth over the scans as shown in Figs. 3a–d and they correspond exactly so we regard our reconstruction (Fig. 3e, h, i) as accurate. We based our reconstructions of the maxilla and lower jaw on the left-hand elements which are less disturbed by skull deformation. The right-hand side does have looser teeth, but this is a result of deformation.

Comments on text

I. 16 ‘oldest currently known’, not ‘current oldest’
amended

I. 18 delete ‘giant’ – the small size of the skull would not allow capturing a titanopteran

amended

I. 151 'quadratojugal foramen' is the generally used term (e.g., Whiteside, 1986)

amended

I. 194 Tetrapods have only one mandible, which comprises two 'mandibular rami' or hemimandibles. Only arthropods have 'mandibles'.

amended

I. 226 There is no such thing as 'lateral mandible'. Do you mean 'lateral surface of the mandibular ramus'?

amended to "lateral surfaces of the mandible"

I. 255-256 The mode of a tooth implantation is very different from other early rhynchocephalians such as *Diphydontosaurus* and *Wirtembergia*, in which the anterior teeth have pleurodont tooth implantation and the posterior ones have acrodont attachment.

We now clarify this, by writing 'The shape and mode of implantation of the teeth support a rhynchocephalian affinity, being primarily acrodont as in *Clevosaurus*, *Sphenodon* and others, even though *Wirtembergia* and *Diphydontosaurus* have pleurodont anterior teeth and acrodont posterior teeth.'

I. 347 Do you mean 'less robust posterior portion of the skull'?

yes, amended

I. 414 There are clear scouring marks on some jaws of *Wirtembergia* – they do not 'appear' to be there.

amended

I.431-432 This sentence is unsupported by a palaeobiogeographical analysis and, in any case, meaningless as Europe did not exist as a geographical entity during the Triassic Period. Please delete this sentence.

amended

Figure 3 is much too dark and will likely be unintelligible in the reduced printed version. 3e has several points that should be fixed: The teeth do not resemble those in the scan images and even those that are not en echelon are too tightly packed together. The squamosal is rendered too small and the quadrate-quadratojugal is incorrectly oriented, both creating an odd-looking suspensorial region. You might want to consult reconstructions of the skulls of other early-diverging rhynchocephalians such as *Diphydontosaurus* (Whiteside, 1986).

As above; corrected.

9th June 2025

Dear Mike

This is the letter I warned you about. Make some fresh coffee. You might need to bring sandwiches.

Your manuscript, "The oldest lepidosaur and origins of lepidosaur feeding adaptations", has now been seen by our referees, and in the light of their advice I am delighted to say that we can in principle offer to publish it. First, however, we would like you to revise your paper to address any remaining points made by the referees, and to make some editorial changes to your paper so that it is as brief as possible and complies with our Guide to Authors. No peer reviewed data should be removed altogether when making these changes.

Many thanks for the final Referee comments and the other information. We respond individually to all the items listed. In that the Referees did not request further changes to the MS, we have made no further changes. We did however check it line by line and are happy with the structure and content of the last version you have. It can be passed to copy-editing.

SOME SPECIFIC ISSUES:

1. Please remove the main figures from the article file and re-supply them individually in an acceptable format such as EPS, AI, PS, PDF, PPT, PSD or XLS (for graphs) with editable vector files. All removed from MS and provided in pdf format. Can provide editable svg too if required.
2. You have duplicate accounts on our content management system, ejp. Please resolve this if you can. If not, just yell. I linked the floating account to my main account.
3. Please provide a supplementary information guide. Our SI guide is on Page 1 of S1, but we can supply this as a separate document if required.
4. There are potential third-party rights issues in the figures - please check sources or if permissions are needed for the fossils, A. helsbypetrae, skull, animal silhouettes etc illustrations in the figures. We checked all figures in main text, extended data and supplement, and find nothing with third-party rights. For example, in Figure 1, all the little skulls are redrafted by us from multiple sources, including generic images without known authorship. In Figure 4, we used silhouettes from phylopic, and we chose their copyright-free option, and cite the artists of all the used images. The other images of the specimen and reconstructions, cladograms, and scan images are by lead author Dan Marke.
5. Please ensure that the text size in all figures is at least 5 pt Arial. All checked.
6. Please re-supply the Extended data figures individually in EPS, JPEG or TIF format. All provided as jpg files; can provide as editable svg as well, if required.
7. For any Supplementary Figures, please check and confirm that:
 - * If data is presented as bar charts, individual data points are shown using overlaid dot plots.
 - * The n number (i.e. the sample size used to derive statistics) is provided and defined as a precise value (not a range), using the wording "n=X samples/cells/independent experiments" etc. where applicable.

- * Any chart axis, error bars, scale bars, symbols and colour scales are defined.
- * Any statistical tests used for data analysis are specified and exact p-values are provided either on the figures themselves, in the legend or in the Source Data file.
- * Wherever representative data such as micrographs are shown, the legend indicates how many times the experiment was repeated with the same results.

None of these issues is relevant to our paper.

TRANSPARENT PEER REVIEW: Nature offers a transparent peer review option for original research manuscripts. We encourage increased transparency in peer review by publishing the reviewer comments and the authors' rebuttal letters if the authors agree. This material is made available as a supplementary peer review file. **Please state in your cover letter either 'I wish to participate in transparent peer review' to opt in, or 'I do not wish to participate in transparent peer review' to opt out.** Failure to state your preference will result in delays in accepting your manuscript for publication. If you wish to opt in to transparent peer review please provide your response to reviewers as a Word file where possible.

I wish to participate in transparent peer review.

Note: We allow redactions to authors' rebuttal and referee comments in the interest of confidentiality. If you are concerned about the release of confidential data, let us know specifically what information you would like to have redacted. We cannot incorporate redactions for any other reasons. Referee names will be published in the peer review file if the referees have signed their comments to authors, or if they explicitly agree to release their name. For more information, see our FAQ page.

No suggestions for redactions.

ORCID--IMPORTANT: All authors identified as 'corresponding author' on the manuscript must have an ORCID associated with their Nature account before submitting the final version of the manuscript. While non-corresponding authors do not have to link their ORCIDs, they are encouraged to do so. Please note that it will NOT be possible to add/modify ORCIDs at the proof stage. Thus, if they wish to have their ORCID added to the paper they must follow the above procedure prior to acceptance. If you have any issues attaching an ORCID identifier to your Nature account, please contact the Platform Support Helpdesk.

I think this is done – we all have ORCID numbers and hope we provided them somewhere. Mike Benton, as Corresponding author, is 0000-0002-4323-1824. David Whiteside is 0000-0003-1619-747X.

In order to avoid delays with publication of your manuscript, please read the guidelines below carefully before resubmission of your manuscript.

STATISTICS: When revising your manuscript, you should ensure that any statistical analysis used is sound and that it conforms to our guidelines. A collection of articles explaining the basics of statistical analysis and advice on how to best present it can be found here. No statistical analyses in the MS.

REPRODUCIBILITY: To ensure that the quality and transparency of methods and

statistical reporting (as discussed here) are sound before the paper is published, we have reviewed your Reporting summary and Editorial policy checklist editorially. I have attached two documents: one listing specific issues related to your manuscript and one containing an annotated version of the Reporting summary. Please ensure that, as well as the more general points below, the points highlighted in the attached documents are addressed in full, both on these forms and within the manuscript. Both forms should be uploaded as a “Related Manuscript” file type. The Reporting summary will be published with your paper. **We revised the two documents and upload the revised versions.**

TITLE: Titles cannot exceed 75 characters (including spaces); they must not contain punctuation. **Title is 67 characters, including spaces.**

SUMMARY PARAGRAPH: Papers start with a fully referenced, bold paragraph, ideally of about 200 words, aimed at readers in other disciplines. Numbers, abbreviations, acronyms or measurements should be avoided unless essential. The summary paragraph consists of 2 to 3 sentences of basic-level introduction to the field; a brief account of the background and rationale of the work; a statement of the main conclusions (introduced by the phrase 'Here we show' or its equivalent); and a conclusion of 2 to 3 sentences putting the main findings into general context so it is clear how the results described in the paper have moved the field forward. A downloadable, annotated example is available here. **This is 195 words, and has been much revised for clarity and logical sequence.**

MAIN TEXT: If further introductory material is necessary, the main text can begin with up to 500 words of introduction expanding on the background to the work (some overlap with the summary is acceptable), before proceeding to a concise, focused account of the findings, and ending with 1 or 2 short paragraphs of discussion. Sections are separated with subheadings (up to 40 characters including spaces) to aid navigation. **We believe we have followed all these suggestions.**

REFERENCES: As a guideline, most papers should include no more than 50 main text references; all additional references can be cited in (and listed after) the Methods section, as detailed below. **We have 46 references for the main text, and 77 in total, when Methods referencing is also included.**

FIGURE LEGENDS: These should be listed sequentially after the main text references and not in the figure files. Each legend should begin with a brief title for the whole figure and continue with a short description of each panel and the symbols used. Legends should not exceed 300 words each. Each figure legend should contain, for each panel where relevant, the following information:

- * the exact sample size (n) for each experimental group/condition, given as a number, not a range;
- * a description of the sample collection allowing the reader to understand whether the samples represent technical or biological replicates (including how many animals, litters, cultures, etc);
- * a statement of how many times the experiment shown was replicated;
- * definitions of statistical methods and measures:

- * very common tests (e.g. t-test, simple Chi-square tests, Wilcoxon and Mann-Whitney tests) can be identified by name only, but more complex techniques should be described in the Methods;
- * whether tests are one-sided or two-sided;
- * whether there are adjustments for multiple comparisons;
- * the statistical test results (e.g., P values);
- * the definition of 'center values' as median or average;
- * the definition of error bars as s.d. or s.e.m.

Descriptions that are too long for the figure legend should be included in the Methods section. **Our figure descriptions follow these suggestions; they do not include any illustrations of statistical tests, and so the mentioned guidelines are not applicable.**

METHODS: The Methods section, which provides the full, step-by-step instructions that would allow other researchers to replicate the results, is included after the main text figure legends. The Methods section will not appear in print but will appear online in the full-text HTML and PDF versions. The Methods section should be written as concisely as possible but should contain all elements necessary to allow interpretation and reproduction of the results. If there are additional references (in the Methods section, Supplementary Information, etc), their numbering should continue from the last entry in the main text reference list, and they should be listed following the Methods section. Specialized methods that require chemical structures, figures, or tables cannot be accommodated in the Methods section of the main text file. If such information is part of the Methods, the entire Methods section must instead be included within a Supplementary Information text file. **We have followed all these suggestions.**

MAIN TEXT STATEMENTS: Several statements (which will not appear in print but will appear online in the full-text HTML and PDF) are required after the Methods (and additional references, if present). First, there should be an Acknowledgements section, listing grant/financial support. Next, we require a detailed Author Contribution statement; the specific contributions of each author, particularly in terms of which authors performed which specific experiments, must be listed. This is followed by a Competing Interest statement. Financial and non-financial interests should be noted here, as well as any patents; patent information should include at a minimum patent number, what is covered by the patent, and who submitted the patent application. Finally, an Additional Information statement should include information regarding reprints and permissions and name the author(s) to whom correspondence and requests for materials should be addressed. Formatting details and an example are available here. **We included all these post-text statements, following the guidelines.**

DATA AND CODE AVAILABILITY STATEMENTS: Any manuscript reporting original research must include a Data Availability statement that makes transparent to the reader the conditions of access to the "minimum dataset" that is necessary to interpret, verify and extend the research in the article. This minimum dataset may be provided through deposition in public community/discipline-specific repositories, custom proprietary repositories (for certain types of datasets), or general repositories like Figshare, Zenodo and Dryad. We strongly discourage providing large datasets in Supplementary Information; the preferred approach is to make data available in

repositories. More information on Nature Portfolio's reporting standards and guidance on preparing your Data Availability statement can be found here. **We provide modest additional data in the Extended data and Supplementary information sections, and provide the original scan data through a doi at the ESRF repository; all detailed in the 'Data Availability Statement'.**

For all studies using custom code or mathematical algorithms that are deemed central to the conclusions, a Code Availability statement must be included, indicating whether and how the code or algorithm can be accessed, including any restrictions to access. The Code Availability statement is listed as a separate section after the Data Availability statement but before any additional references. Code should be deposited in a DOI-minting repository such as Zenodo, Gigantum or Code Ocean and cited in the reference list. Authors are encouraged to manage subsequent code versions and to use a license approved by the open source initiative. Additional details can be found here. **Not relevant to our paper.**

DISPLAY ITEMS: We suggest that you take stock of all data that have been generated throughout the review process and ensure that only the data most central to the conclusions are presented in the main text figures. Any figures included within the main text file during the review process must be removed from the final main text file and uploaded as separate, individual files; they will be integrated into the main paper in print and online. An overview of the key features of this presentation may be found here. **We have shuttled material between core Display items and Extended data, and believe we have the right distribution.**

Figures should be comprehensible to readers in other disciplines and assist in understanding of the paper. Main text figures (but **not** Extended Data) must be provided in production-quality versions in an editable format (i.e., .ai, .cmx, .cdr, .doc, .eps, .pdf, .ppt, .ps, .psd, .svg and .xls); we cannot accept figures in .cvs, .gif, .jpg, .png and .tif formats. We highly encourage you to consult our artwork guidelines. They should be as small and simple as is compatible with clarity. All panels of a figure should be logically connected and assembled on a single page in a rectangular shape; any essential alignments (parts horizontal, vertical, spacings, etc) should be indicated. Each panel of a multipart figure should be sized so that the whole figure can be proportionally reduced and reproduced on the printed page at the smallest size at which essential details are visible. Nature's standard figure sizes are either 9 or 18 cm wide; the maximum permitted height is 17 cm. Panels should be arranged to fit these widths while minimizing excess space around the panels. Tables should be prepared using the Table menu in Word. As we must be able to edit the figures so that they conform to our house style, the submission of files that are incorrectly formatted, flattened, or of insufficient resolution may delay final acceptance of your manuscript. **Figures have been designed for accuracy, clarity, accessibility, etc, and following the Nature guidelines.**

THIRD PARTY RIGHTS: You must provide proof that you have secured permission to use any third party materials that appear in any part of your manuscript, including Extended Data and Supplementary Information. Please fill out a Third Party Rights Table, and upload this with the final version of your manuscript. Third party materials include any

figures, tables, images, videos or text boxes that are reproductions or adaptations of items that have previously been published elsewhere and/or are owned by a third party. This includes pictures taken by professional photographers, maps and images downloaded from the internet. You will need to obtain the right to use each of these items before your paper can be accepted for publication. You will also need to give proper attribution to the copyright holders in your paper. Please ensure you upload any necessary grants of rights alongside the final version of your manuscript. More information is available on our Rights and permissions page. Failure to obtain the appropriate rights and to supply a completed third party rights table will delay the publication of your article. The editorial assistant (cc'd) can help with any questions. **All images (specimen photographs, digital segmented images, cladograms) are by leading author, Dan Marke. We have used only copyright-free silhouettes in Figure 4, and these are acknowledged to source in the caption.**

COVER ARTWORK: We welcome submissions of artwork for consideration for our cover. More information can be found in our guide for cover artwork. The file name(s) should include the manuscript reference number and be labelled as a cover suggestion; a short description is also preferred. Illustrations should be selected more for their aesthetic appeal than for their scientific content. We cannot promise that your suggestions will be selected for the cover, as competition is intense. **We have commissioned a painting from Bob Nicholls, intended for cover art, and this should be available by July 5th – we will get back to you then, but hope the paper can be held until that image is available. Apologies for this delay, but Nicholls, as one of the leading palaeoartists in the world, has heavy commissions.**

IMAGE INTEGRITY: We strongly advise that you go carefully through all the data (including Extended Data and Supplementary Information) to ensure there are no accidental image/data duplications, other image manipulations or data errors. Such issues generally require correction after publication. Any image provided for publication, either in print or online (including Extended Data and Supplemental Information), may be subject to a quality control process to check for image integrity and manipulation. A discussion of our standards regarding how images should be prepared and presented can be found here. **We believe all images are unique, required, and meet *Nature* guidelines.**

EXTENDED DATA: Extended Data do not appear in print but are included online within the full-text HTML and integrated in the downloadable PDF. Extended Data are an integral part of the paper, and only data that directly contribute to the main message should be included. All Extended Data must be referred to in the main text, and their legends should be listed sequentially at the end of the main text file, not in the Extended Data files. Extended Data should be assembled into a maximum of 10 A4 size, multi-panelled display items. They must be supplied as individual files in .jpg, .tif or .eps format **only**. They should be of the same quality as the main figures, but there are important differences in their formatting. More specific instructions are provided here. If you need to describe a complex process, we encourage you to add a schematic of the main finding as part of the Extended Data to aid readers unfamiliar with the immediate discipline. **We provide all Extended data images as separate files, as well as a**

compilation pdf to show them with captions.

SUPPLEMENTARY INFORMATION: Supplementary Information (SI) is online-only, peer-reviewed material that is essential background to the study (e.g., large data sets, more complex methods, and calculations), but which is too large or impractical, or of interest only to a few specialists, to justify inclusion in the print version of the paper (see here for further details). While SI should not typically contain data figures (any figures additional to those appearing in the main text should be formatted as Extended Data), we require that the raw, uncropped data for gels be presented as an SI figure (see below). Tables may be included in SI, but only if they are unsuitable for formatting as Extended Data (e.g., tables containing large data sets that cannot fit a single page or raw data tables that are best suited to Excel files). If a manuscript has SI, each discrete SI item (e.g., videos, tables) must be referred to at an appropriate point in the main text file. You must also provide a Word file entitled "SI Guide", containing a cover page with manuscript title and author information; a table of contents (preferably with page numbers); and then any SI text, notes, figures, and titles and legends for any separate SI files; for additional information see here. **We provide an SI Guide, as recommended.**

We recommend that you pay careful attention to the formatting of the SI because it is not subedited. After the paper has been accepted, SI files can only be amended for critical changes to the scientific content, not for style. **Understood and agreed.**

SOURCE DATA (GRAPHS): To increase transparency, we strongly encourage you to provide, in spreadsheet form, the data underlying the graphical representations used in figures. For all experiments presenting data from animal models, this is a requirement and is not optional. This is in addition to our well-established data-deposition policy for specific types of experiments and large datasets. Online readers of the manuscript will be able to access the graphical source data directly from the figure legend. Spreadsheets must be submitted in .xls, .xlsx or .csv formats. One file per figure is permitted. If there is a multi-panelled figure, the source data for each panel should be clearly labeled in the file; alternatively the source data for a figure can be included in multiple, clearly labeled sheets within an Excel file. File sizes of up to 30 MB are permitted, but it is expected that the vast majority of graphical source data files will be considerably smaller than this. When submitting these files with your manuscript, you should select the "Source Data" file type and use the title field in the file description tab to indicate the figure(s) to which the source data pertain. Source data should not be provided as Extended Data. **We do not have any graphs.**

DATA DEPOSITION: The following specific points may be relevant to your paper, so please ensure that you provide the following information:

* Papers containing new or revised formal taxonomic nomenclature for animals, whether living or extinct, are accepted conditional on the provision of LSIDs (Life Science Identifiers) by means of registration of such nomenclature with ZooBank, the online registration system for the International Code of Zoological Nomenclature (ICZN). ZooBank LSIDs can be resolved and the associated information viewed through any standard web browser by appending the LSID to the prefix "<http://zoobank.org/>".

* We strongly encourage deposition of 3D morphological data in a suitable repository such as MorphoBank, MorphoSource or similar; the relevant accession numbers should be listed in the Data Availability statement.

We will not send your revised paper for further review. If the revised paper is in our format (as detailed above), in accessible style and of appropriate length, we shall begin the acceptance process.

In order to accept your paper, we require the following electronic files:

* A cover letter describing your response to any editorial comments and detailing any format changes during revision, particularly if the overall length is affected. **This is it.**

* A point-by-point response (preferably in Word) to any remaining issues raised by our referees. **And this is also it.**

* The final version of your text as a Word document. Word Equation Editor/MathType should be used only for formulae that cannot be produced using normal text or symbol font. If this is not possible, the manuscript can be supplied as a single plain vanilla TeX or LaTeX file that includes all references and abbreviations, with no special formatting, as well as a PDF version that is uploaded as a 'related manuscript file'. **Provided**

* Production-quality versions of all figures (see above). **Provided**

* The final version of the Extended Data. **Provided**

* The final version of any Supplementary Information, presented as one file (ideally a PDF) if feasible, as well as a separate SI Guide. **Provided**

* Source Data, if appropriate. **Not appropriate.**

* For optimal quality videos we encourage H.264 encoding and a standard aspect ratio of 16:9 (4:3 is second best), without compression. **Not relevant.**

* Completed and signed copies of the following **five (or six) forms**, uploaded as a "Related Manuscript File" file type:

- 1) Biology editorial checklist;
- 2) Manuscript checklist;
- 3) Reporting summary;
- 4) Editorial policy checklist;
- 5) Third-party rights table;
- 6) Code and software submission checklist (if applicable).

All relevant documents provided.

Nature has now transitioned to a unified Rights Collection system which will allow our

author services team to quickly and easily collect the rights and permissions required to publish your work. Once your paper is accepted, you will receive an email in approximately 10 business days providing you with a link to complete the grant of rights. If you choose to publish Open Access, our author services team will also be in touch at that time regarding any additional information that may be required to arrange payment for your article. If you have any questions please contact asjournals@springernature.com. **Understood.**

You may need to take specific actions to achieve compliance with funder and institutional open access mandates. If your research is supported by a funder that requires immediate open access (e.g. according to Plan S principles) then you should select the gold OA route, and we will direct you to the compliant route where possible. If you select the subscription publication route our standard licensing terms will need to be accepted, including our self-archiving policies. Those standard licensing terms will supersede any other terms that you or any third party may assert apply to any version of the manuscript. **Understood.**

We hope to hear from you within two weeks; please let us know if the process may take longer.

Yours sincerely

Henry

Dr Henry Gee

Senior Editor Biological Sciences

nature@nature.com

<http://www.nature.com/nature>

See Nature's new guide to authors at www.nature.com/nature/submit

Referees' comments:

Referee #1 (Remarks to the Author):

Many thanks for the opportunity to review the revised version of this very interesting manuscript. I am pleased to see that the authors have made every effort to address my initial suggestions and concerns.

I still question the absence of palatal teeth, but I thank the authors for providing further clarification on their original reasoning. I accept they have made as good a case as they can with the provision of new scans and some clarifications around the absence of tooth wear facets. This is very helpful and it does strengthen their case. While I still consider the presence or absence of palatal teeth to be a key part of the manuscript, the support the authors have provided for their point of view is reasonable.

Many thanks; no action.

Referee #2 (Remarks to the Author):

Thank you for fully addressing the comments from my previous review.
Many thanks; no action.